# Type-H endothelial cell protein Clec14a orchestrates osteoblast activity during trabecular bone formation and patterning
Georgiana Neag [1], Jonathan Lewis[1], Jason D. Turner [1], Julia E. Manning[1], Isaac Dean [2], Melissa Finlay[1], Gowsihan Poologasundarampillai [3], Jonathan Woods[1], Muhammad Arham Sahu [1], Kabir A. Khan[2,4,5], Jenefa Begum[1], Helen M. McGettrick [1], Ilaria Bellantuono[6], Victoria Heath[2], Simon W. Jones [1], Christopher D. Buckley[1,7], Roy Bicknell[2] & Amy J. Naylor [1] ✉

Type-H capillary endothelial cells control bone formation during embryogenesis and postnatal growth but few signalling mechanisms underpinning this influence have been characterised. Here, we identify a highly expressed type-H endothelial cell protein, Clec14a, and explore its role in coordinating osteoblast activity. Expression of Clec14a and its ligand, Mmrn2 are high in murine type-H endothelial cells but absent from osteoblasts. $Clec14a^{-/-}$ mice have premature condensation of the type-H vasculature and expanded distribution of osteoblasts and bone matrix, increased long-bone length and bone density indicative of accelerated skeletal development, and enhanced osteoblast maturation. Antibody-mediated blockade of the Clec14a-Mmrn2 interaction recapitulates the $Clec14a^{-/-}$ phenotype. Endothelial cell expression of Clec14a regulates osteoblast maturation and mineralisation activity during postnatal bone development in mice. This finding underscores the importance of type-H capillary control of osteoblast activity in bone formation and identifies a novel mechanism that mediates this vital cellular crosstalk.

Angiogenesis, the process by which new blood vasculature sprouts from existing vessels, is a pivotal mechanism in embryonic and postnatal organ development and is critical to the maintenance of homeostasis throughout life[1-3]. Recently, there has been renewed interest in the role of bone angiogenesis in the regulation of postnatal skeletal growth, bone remodelling and bone repair[4-7]. Technological advances in microscopy, coupled with the development of novel bone processing techniques[8,9], have provided the ability to investigate the vascular network of the bone in 2D and 3D, leading to the discovery of new bone capillary subtypes, such as the type-H capillary[5]. Described as active modulators of osteogenesis[5,8], type-H capillaries expand from the metaphysis into the growth plate during embryogenesis[9] and postnatal bone growth[5,8]. Type-H capillary endothelial cells provide guidance and molecular signals to osteoblasts migrating along their abluminal side[5,9-11]. It is now clear that close co-dependent interactions between osteoblasts and endothelial cells (EC) are essential in controlling

trabecular bone mass during embryogenesis and postnatal bone formation[10,12-18].

Identification and characterisation of novel targetable pathways to modulate bone angiogenesis and to selectively increase bone formation reflects a current need to identify novel therapies to boost bone formation[19-21]. The recently identified interactome between CD248-MMRN2-CLEC14A, represents one such pathway[22]. Multimerin 2 (MMRN2) is an endothelial cell-specific extracellular matrix (ECM) protein that can concomitantly bind two C-type lectin domain (CTLD) 14 family members: CLEC14A, which is expressed by endothelial cells, and CD248, which is expressed on osteoblasts and other stromal cells, such as pericytes and fibroblasts[23-27]. We, and others, have shown that both Clec14a and Cd248 are significant regulators of embryonic and pathologic angiogenesis, cell adhesion and migration[24,25,28,29]. In addition, we have previously demonstrated that Cd248 acts as a negative regulator of osteoblast miner-

[1]Rheumatology Research Group, School of Infection, Inflammation and Immunology, College of Medicine and Health, University of Birmingham, Birmingham, UK. [2]School of Medical Sciences, University of Birmingham, Birmingham, UK. [3]School of Health Sciences, College of Medicine and Health, University of Birmingham, Birmingham, UK. [4]Department of Medical Biophysics, University of Toronto, Toronto, ON, Canada. [5]Biological Sciences Platform, Sunnybrook Research Institute, Toronto, ON, Canada. [6]Healthy Lifespan Institute, School of Medicine and Population Health, University of Sheffield, Sheffield, UK. [7]The Kennedy Institute of Rheumatology, University of Oxford, Oxford, UK. ✉e-mail: a.naylor@bham.ac.uk

alisation, with genetic deletion of *Cd248* resulting in increased osteoblast-driven mineralisation both in vitro and in vivo[30]. Moreover, recent research has also identified upregulation of Clec14a expression during bone healing[31]. In the context of these findings, we explored the role of Clec14a in bone formation and show that Clec14a and Mmrn2 are highly expressed by type-H endothelial cells and that Clec14a acts as a negative regulator of bone formation during postnatal skeletal growth in the mouse.

## Results

### Clec14a and Mmrn2 are highly expressed in type-H vessels

To understand their potential for a role in angiogenesis and bone accrual, Clec14a and Mmrn2 protein expression was assessed using immuno-fluorescence microscopy in the murine long bone. This analysis confirmed the expression of both proteins in the bone metaphysis (Fig. 1). Due to its high sensitivity to shear stress, Clec14a is rapidly upregulated upon cessation of circulation[29]. To ensure that the expression of Clec14a is present in the metaphysis in the living animal, and not artificially upregulated upon sacrifice, it was immunolabelled in vivo by systemic injection of biotinylated anti-Clec14a antibody. The resultant binding was detected with APC-conjugated streptavidin, demonstrating localisation predominantly to the bone metaphysis type-H vessel endothelium (Fig. 1A), with particularly high expression noted at the anastomotic arches and bulbs (Fig. 1A i, ii). Mmrn2 is a secreted, extracellular matrix protein and the staining pattern observed matched that expected if produced by type-H vessel endothelial cells, which postnatally are primarily confined to the metaphysis[5,8] (Fig. 1B). Reduced, more punctate expression of Mmrn2 was observed in the dia-physis bone region where bone marrow-regulating L-type (Emcn^low) vessels reside[5]. In addition to protein expression, a publicly available single-cell RNA-sequencing data (dataset: GSE128423) was mined to identify *Clec14a* and *Mmrn2* mRNA expression levels on cells isolated from the bones of 8-week-old C57BL/6 J mice (Fig. 1C). *Emcn*, *Clec14a* and *Mmrn2* expression is confined to endothelial cell (EC) clusters, whilst *Pecam1* (*Cd31*) is also expressed by a minority of MSC and chondrocytes. Additional analysis of publicly available bulk RNA-seq from flow cytometry-sorted cells (E-MTAB-4066 dataset) and single-cell RNA-seq (GSE128423 dataset) data demonstrated that both *Clec14a* and *Mmrn2* are amongst the most differentially upregulated genes in type-H compared to type L endothelial cells (Supplementary Fig. 1A–E). Cumulatively, these data confirm that within the bone, expression of both Clec14a and its ligand Mmrn2 is confined to EC, with pronounced expression in type-H vessel EC.

### Germline *Clec14a* deletion impacts type-H capillary patterning

Given the positive association of type-H capillary density to bone mass[8,12], the area covered by these capillaries was analysed to assess the impact of deletion of Clec14a on bone vascularisation. The fraction of Cd31^high Emcn^high type-H EC was quantified at the peak of their ratio expression compared to Cd31^lo Emcn^lo type-L EC in the postnatal long bone using flow cytometry (Fig. 2A, B). A significant decrease in type-H endothelium percentage was identified in 2-week-old *Clec14a^−/−* mice compared to *Clec14a^+/+* (wildtype - WT), whilst no difference in L-type endothelium (Cd31^low Emcn^low) was detected (Fig. 2B). The number of capillaries in both *Clec14a^+/+* and *Clec14a^−/−* samples increased over time from postnatal day (P)4 to 4-weeks, in positive correlation with the increase in bone diameter and no significant differences were observed in vessel density between *Clec14a^+/+* and *Clec14a^−/−* (Supplementary Fig. 2A–D). TUNEL staining was employed to identify whether apoptosis rates were equivalent in *Clec14a^+/+* and *^−/−* but no apoptosis was detectable at the 4-week timepoint (Supplementary Fig. 2E).

Immunofluorescence imaging of tibiae from *Clec14a^−/−* and *Clec14a^+/+* mice was performed to assess the organisation of the vasculature. In mice of both genotypes at P4, immunofluorescence imaging of the tibia presented a heavily interconnected vascular bed (as detected by anti-Emcn antibody staining) with an approximately equal coverage of type-H and -L capillaries (Supplementary Fig. 3). In the juvenile mouse (4-week-old), the type-H bed was diminished in both genotypes by comparison to the neonates (P4) and

became distinctly localised to the superior part of the metaphysis (Fig. 2C). A decrease in the length of type-H columns and a corresponding decrease in the area covered by the type-H column vascular front in the metaphysis was detected in *Clec14a^−/−* samples at 4-weeks (Fig. 2D) but not at P4 (Supplementary Fig. 4A–E). No differences were identified in the directional orientation of type-H capillaries in the metaphysis relative to the direction of bone growth (Supplementary Fig. 4F–H).

As previously reported[10], at the leading edge of type-H vessels, bud-like vessel protrusions are visible and anastomosis of those protrusions by arches are observed. By comparison to controls, *Clec14a^−/−* type-H bulges formed an invading vessel front line which appeared more disorganised than in the *Clec14a^+/+*, with type-H bulges advancing unevenly (dotted areas Fig. 2E). In addition, blunting of H-vessel buds (Fig. 2F, G) and reduced EC filipodia formation (Fig. 2H) was detected in *Clec14a^−/−* images at 4-weeks but was not observed in sections from P4 *Clec14a^−/−* mice (Supplementary Fig. 3).

To establish whether organisation within the growth plate was affected by genetic deletion of Clec14a, staining of glycosaminoglycans was employed to highlight proliferating chondrocyte columns and hypertrophic regions (Fig. 2I, J). No overt differences were detected between genotypes.

### Clec14a controls osteoblast distribution during development

To understand whether endothelial cell-specific Clec14a influences osteo-blast migration and differentiation during bone development, we observed the effect of *Clec14a* genetic deletion on the localisation of immature and mature osteoblasts via immunostaining for Osterix (Osx) and Type-I Collagen (Col1a1), respectively (Fig. 3).

Regardless of genotype, Osx^+ osteoblasts were present in large numbers in the tibial metaphysis (mp) and in the endosteum, with a relatively lower number of osteoblasts observed within the diaphysis (dp) (Fig. 3A). Osteoblasts were localised near type-H vessels, except surrounding the bud-like protrusions, at the leading edge of vessels, where osteoblasts were notably absent (Supplementary Fig. 5). Osteoblasts were also observed lining avascular bone pits in the inferior metaphysis and caudally towards the diaphysis, with lower cellularity compared to the rest of the bone (Supplementary Fig. 6A, B). Endosteal osteoblasts were tightly arranged in columns on either side of the interior part of the bone cortex, with one column lining the medial side of the cortex next to the endosteum and one column lining the lateral side of the cortex towards the periosteum. A limited number of Osx^+ cells were also detected in the chondrocyte-populated area of the growth plate (Supplementary Fig. 5), as has been previously observed[4]. To assess whether proximity of osteoblasts to endothelial cells was affected by the absence of Clec14a, distance between cells was quantified in both the metaphysis (type-H) and diaphysis (type-L) (Supplementary Fig. 6C). Osteoblasts were on average located further from type-L vessels than type H, but no differences were observed between the genotypes.

*Clec14a^−/−* neonatal mice showed a significant increase compared to *Clec14a^+/+* in immature Osx^+ osteoblast number in the inferior metaphysis towards the diaphysis (terminology simplified to diaphysis in all images in Fig. 3) at P4, but not in the upper metaphysis (Fig. 3A, B). No difference was detected in the number of proliferating Osx^+ cells (Supplementary Fig. 6D). Furthermore, an increased length of osteoblast chords formed on the lining surface of newly forming trabeculae which are longer (extend further from the growth plate) compared to Clec14a^+/+ controls was observed (Fig. 3C).

To further probe the maturity status and activity of osteoblasts, tibial sections were immuno-stained against Type I Collagen (Col1a1). Intracel-lular and extracellular staining of Col1a1 was identified throughout the bone, with focal points in the endosteum and on the edges of bone trabe-culae in the inferior region of the metaphysis and in the diaphysis (Fig. 3D). No changes were observed in the area covered by collagen in the two genotypes (Fig. 3E). The mean length of Col1a1^+ chords was found to be longer in *Clec14a^−/−* tibiae as compared to *Clec14a^+/+* controls (Fig. 3F).

### *Clec14a^−/−* mice have increased bone density

To assess the effect of *Clec14a* deletion on bone growth, body weight and tibial bone length were measured in 2-, 4-, 8- and 30-week-old Clec14a^+/+

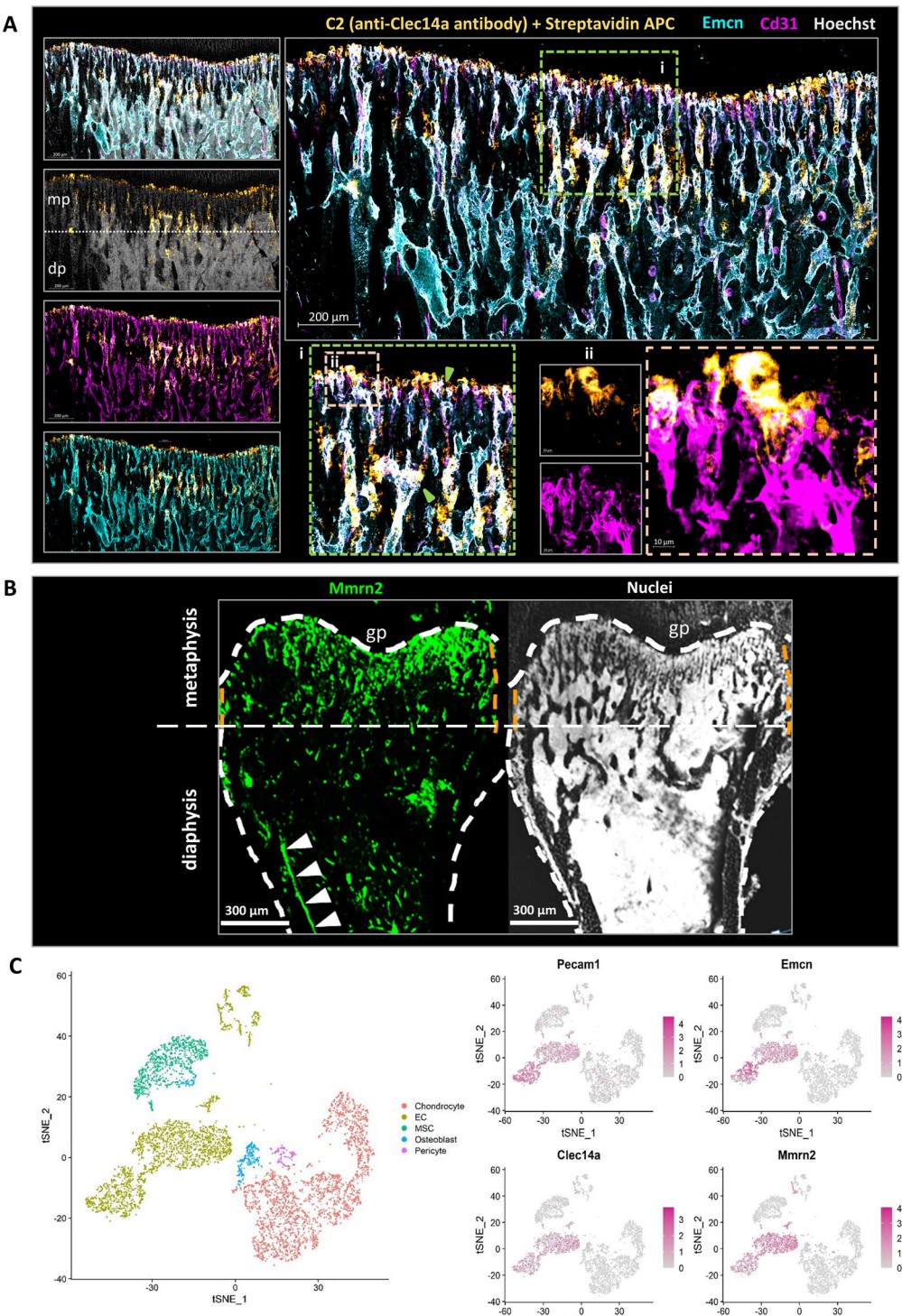

**Fig. 1 | Mmrn2 and Clec14a expression at sites of trabecular bone formation.**
**A** Maximum intensity projection images show immunofluorescent staining of Clec14a (orange), Endomucin (Emcn - cyan), Cd31 (magenta), Hoechst (grey). Clec14a expression was tracked in vivo, a biotin-labelled monoclonal antibody (C2) raised against Clec14a was injected intraperitoneally and tissue was collected 24 h post injection. Scale bar is 200 μm. **i**. Region of interest with arrowheads (green) indicating areas where co-expression of all markers (white) is evident. **ii**. High magnification image demonstrating high expression of Clec14a within bud-like vessel protrusions and anastomotic arches. Scale bar is 10 μm. **B** Maximum intensity projection images show immunostaining of Mmrn2 (left image - green) and nuclei (Hoechst, right image - grey). Mmrn2 is located predominantly in the anatomical

area normally covered by type-H vessels in the metaphysis (mp), moderate Mmrn2 expression is also observed in the region where type-L vessels reside (dp). White arrowheads indicate Mmrn2 staining on an arterial vessel (arterial vessel identified with reference to Kusumbe et al. 2014, Fig. 1 and extended data Fig. 2). White dashed line demarcates the growth plate metaphysis border and the periosteum. Abbreviations: gp-growth plate, mp-metaphysis, dp-diaphysis. Scale bars are 300 μm.
**C** Expression of *Clec14a* and *Mmrn2* in the bone stroma of adult mice (GSE128423). Left panel: t-SNE map of the stromal clusters in the bone coloured-coded according to their cluster name. Right panel: t-SNE maps reveal expression of H-type endothelial cell (EC) markers *Pecam1* and *Emcn* alongside *Clec14a* and *Mmrn2*.

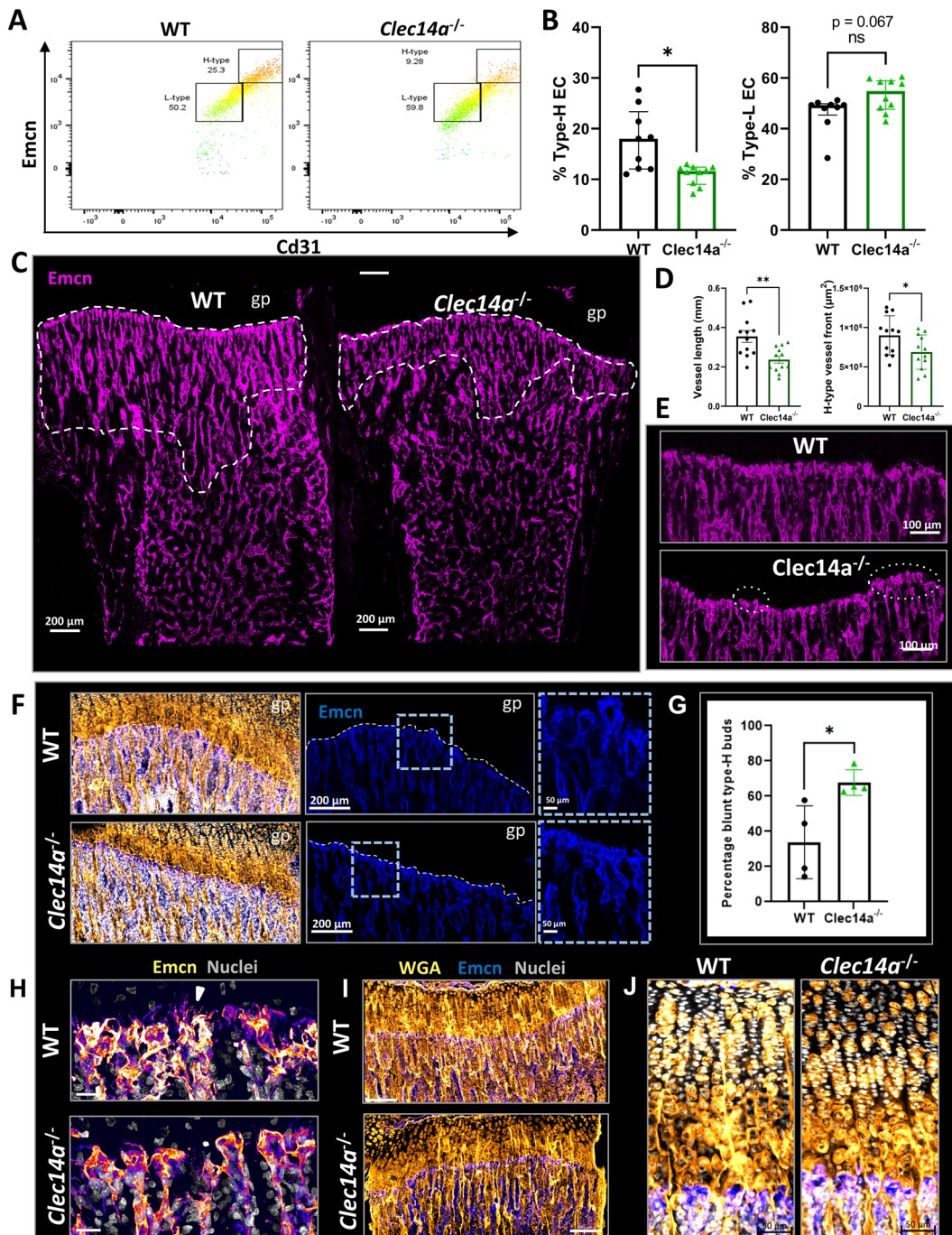

**Fig. 2 | Changes in vascularisation of the long bone in *Clec14a*$^{-/-}$ mice.**
**A** Representative flow-cytometry heatmap scatter plots showing Cd31$^{high}$ Emcn$^{high}$ EC (H-type) and Cd31$^{low}$ Emcn$^{low}$ (L-type) endothelial cells (EC) in Cd45$^-$ Ter119$^-$ cells isolated from the long bones of *Clec14a*$^{+/+}$ and *Clec14a*$^{-/-}$ 2-week-old pups. **B** Relative abundance of H-type compared to L-type EC in 2-week-old mouse tibiae from *Clec14a*$^{-/-}$ mice compared to *Clec14a*$^{+/+}$ (WT). Data was analysed with a Mann–Whitney test, results are presented as median ± IQR, *$P < 0.05$, ns = not significant. **C** High resolution 3D confocal microscopy of the juvenile (4 weeks old) *Clec14a*$^{+/+}$ (WT) and *Clec14a*$^{-/-}$ mouse tibia capillary network. The panel shows representative maximum intensity projection images depicting organisation of H-type vessels in *Clec14a*$^{+/+}$ (WT) and *Clec14a*$^{-/-}$ tibia. Vessels in the tibia section have been immunolabelled with Emcn (magenta). White line marks the limits of the type-H vessel front, note the shorter H capillary front in the *Clec14a*$^{-/-}$ images. Scale bars are 200 μm. **D** Left graph: quantification of the length of H type columns in the metaphysis, data represents mean values for each biological replicate. Right graph: quantitative evaluation of the length of the type-H vessels front. In **D** and **E**, data is represented as mean ± SD, results were analysed by unpaired T test. **P < 0.01;

*$P < 0.05$; ns = non-significant. **E** Representative enlarged images of the leading type-H vascular front (Emcn, magenta) in *Clec14a*$^{+/+}$ and *Clec14a*$^{-/-}$ mice. Note dysregulation of the advancing H-type bulges protruding into the growth plate (highlighted by the white dashed ovals in the *Clec14a*$^{-/-}$ image). Scale bars are 100 μm. **F** Representative immunofluorescent images of type-H vessels labelled with Emcn (blue) and counterstaining of cells and extracellular matrix in the metaphysis and growth plate with wheat germ agglutinin (WGA - orange). Note blunting of type-H vessel buds in *Clec14a*$^{-/-}$. Scale bars are 200 μm (main images) and 50 μm (high magnification insets). **G** Quantitative evaluation of the fraction of type-H vessels buds showing blunted phenotype in the bone metaphysis. **H** Maximum intensity projection images show immunostaining of Emcn (fire – colour gradient with high expression in orange and low expression in magenta) and nuclei (Hoechst, grey) in sections collected from 4-week-old *Clec14a*$^{+/+}$ and *Clec14a*$^{-/-}$ male mice. Note absence of filopodia (white arrowhead) in *Clec14a*$^{-/-}$ images. Scale bars are 20 μm. **I, J** Immunofluorescent images of the chondro-osseous interface (wheat germ agglutinin WGA - orange, Emcn - blue). Note the normal patterning of the growth plate in both *Clec14a*$^{+/+}$ and *Clec14a*$^{-/-}$. Scale bar = 200 μm (**I**) and 50 μm (**J**).

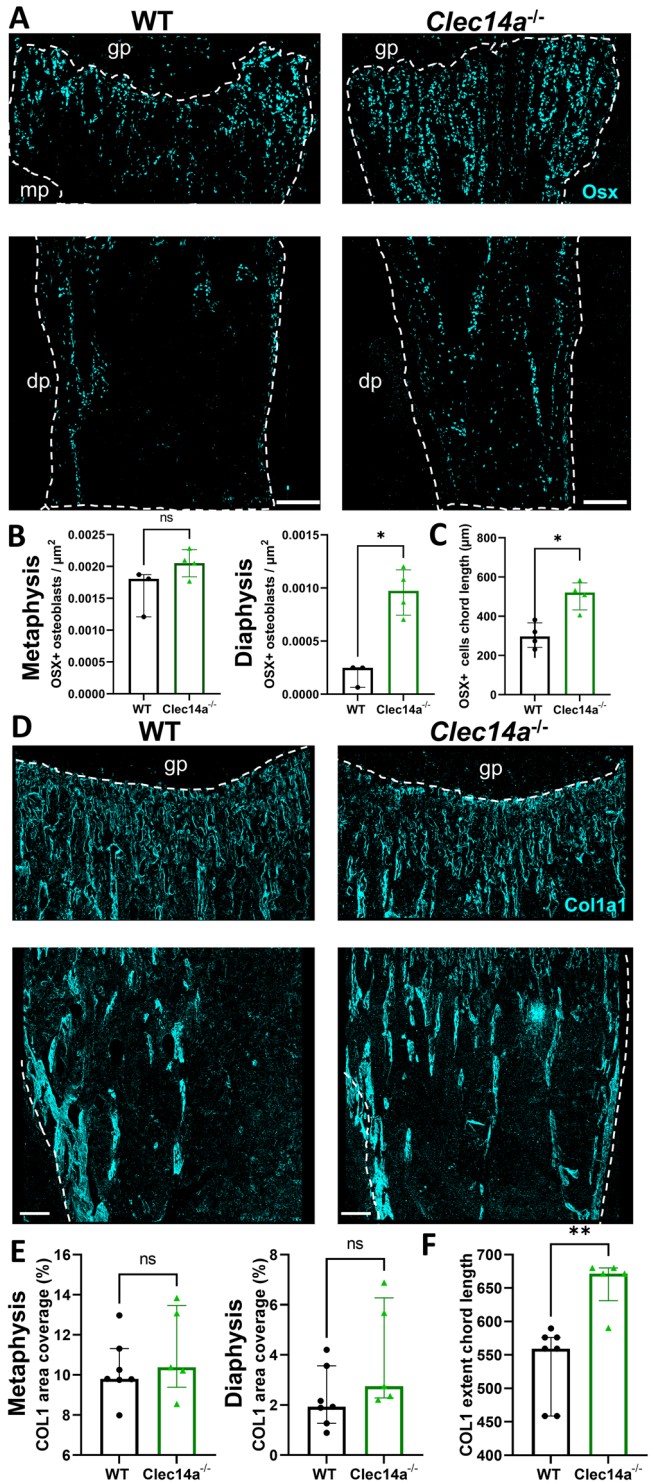

**Fig. 3 | Increased extension of the osteoblast front in the long bone of *Clec14a⁻/⁻* mice.** **A** Representative images of Osx (cyan) immunolabeled metaphyseal and diaphyseal bone sections from P4 neonatal pups. Dashed line demarcates the bone edge of images and the most distal point of the proximal growth plate. Note the extension of Osx⁺ cells in *Clec14a⁻/⁻*. Scale bars are 100 μm. **B** Quantitative measurements of osteoblast density per μm² in the P4 mouse tibia metaphysis and diaphysis. **C** Quantification of the length of osteoblast chords in the neonatal (P4) murine tibia. An osteoblast chord was defined as an aggregation of Osx⁺ cells organised in a column. The length of osteoblast chords from the proximal end of the metaphysis to their most distal point was measured. Individual dots represent the mean length of osteoblast chords in one biological replicate. Data was analysed with a Mann–Whitney test, results are presented as median ± IQR *$P < 0.05$, ns = not significant. **D** Representative maximum intensity projections of Col1a1 immunolabeled bone sections in 4-week-old mice. Upper image: white dashed line delineates the boundary between the growth plate (avascular) and the metaphysis. Lower image: dashed line demarcates the area covered by bone including the bone endosteum and periosteum. Scale bars are 100 μm. **E** Quantification of the percentage (%) area covered by Col1a1 immunolabeling in the tibial metaphysis (left) and diaphysis (right). **F** Quantification of the length of type I collagen chords in the juvenile murine tibia. Data analysed with a Mann–Whitney test, results are presented as median ± IQR, **$P < 0.01$, ns = not significant.

The effect of *Clec14a* genetic deletion on long bone development at earlier developmental timepoints was also assessed. Bone elongation of embryonic day E14.5 metatarsal explant cultures was observed over 7 days and a significant increase in *Clec14a⁻/⁻* explant length was observed at all timepoints (Fig. 4E, F).

To investigate the quantity and spatial organisation of bone, μCT analysis of trabecular (Fig. 4G,H) and cortical bone (Supplementary Fig. 7A, B) in the tibia were performed. Strikingly different spatial organisation of trabecular bone was observed in the *Clec14a⁻/⁻* male mice compared to *Clec14a⁺/⁺* controls (Fig. 4H) with evidence of a substantial increase in percentage bone volume (BV/TV %) and trabecular number (Tb.N) in the metaphysis of *Clec14a⁻/⁻* male mice at 4-weeks and 8-weeks of age, which had normalised by the later timepoints. These increases in trabecular bone mass corresponded to a significant decrease in trabecular separation at these timepoints (Fig. 4H). Trabecular bone extent was quantified by measuring the distance from the metaphyseal growth plate to the most distal site where trabecular bone was identified (Fig. 4I), which was pronouncedly increased in *Clec14a⁻/⁻* at all stages compared to *Clec14a⁺/⁺* controls (Fig. 4J). The same trends towards increased bone density and extent were seen in tibiae from female *Clec14a⁻/⁻* mice compared to *Clec14a⁺/⁺* at 4 and 8 weeks of age but did not reach statistical significance (Supplementary Fig. 8). Increases in bone area and tissue area were identified in the *Clec14a⁻/⁻* compared to *Clec14a⁺/⁺*, consistent with their larger size, but no significant differences in the architecture of cortical bone or in its biomechanical parameters in response to 3-point bending were detected (Supplementary Fig. 7).

### *Clec14a⁻/⁻* osteoblasts show accelerated maturation

To better understand molecular alterations at the gene transcription level in response to the absence of Clec14a, the transcriptomic signature of *Clec14a⁻/⁻* mice was investigated using bulk RNA sequencing of neonatal calvarial-cell isolates. Unsupervised hierarchical clustering and principle-component (PC) analysis showed that the samples clustered by genotype (Fig. 5A, B) and a total of 1223 genes were identified as differentially expressed (~2% of the total genes detected), of which 522 were upregulated and 701 were downregulated in *Clec14a⁻/⁻* compared to *Clec14a⁺/⁺* (Fig. 5C).

Gene set enrichment analysis (GSEA) demonstrated that *Clec14a⁻/⁻* samples strongly expressed markers enriched in pathways related to endochondral ossification (Fig. 5D, E) and osteoblast proliferation and maturation (Supplementary Fig. 9), which include bone extracellular matrix genes (*Col1a1*, *Col1a2*)[32–34], matrix mineralisation driver alkaline phosphatase (*Alpl*)[33] and nucleator of hydroxyapatite crystal formation bone sialoprotein (*Ibsp*)[34–36] (Supplementary Fig. 9). A significant increase in

and *Clec14a⁻/⁻* male mice (Fig. 4A). No major differences in body weight were observed at any time point although a subtle increase in weight at 4-weeks was detected (Fig. 4B). Total tibia length (Fig. 4C) and the distance between the proximal growth plate and the tibial-fibular junction (Fig. 4D) were measured. In both genotypes (*Clec14a⁺/⁺* and *Clec14a⁻/⁻*), the most active period of bone elongation occurred between 2- and 4-weeks, where tibiae doubled in length. Both measurement methods revealed similar findings, that bone elongation occurs more rapidly in the absence of Clec14a during postnatal bone development (Fig. 4C, D). The most striking difference was observed at 2-weeks of age where tibiae in *Clec14a⁺/⁺* controls were markedly shorter than *Clec14a⁻/⁻* (Fig. 4C).

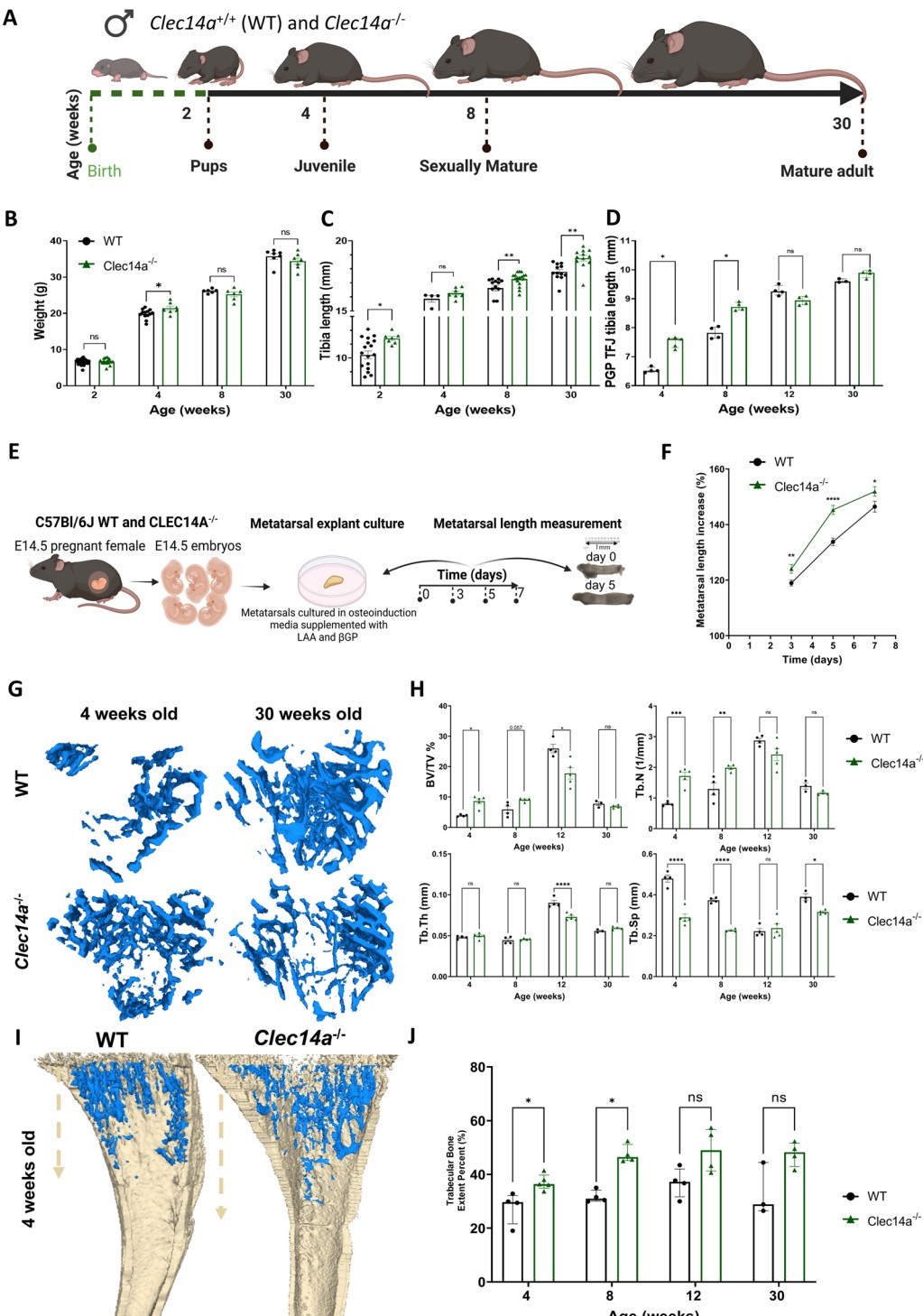

**Fig. 4 | Bone morphometric parameters in *Clec14a⁻/⁻* male mouse tibia. A)** Diagram illustrating morphometry analyses timepoints (generated in BioRender): Tibiae were collected from C57BL/6J male mice at different skeletal maturation stages: juvenile (2- and 4-weeks-old), sexually mature (8-weeks-old) and mature adult (30-week-old). **B** Body weight. **C** Total tibiae length and **D** tibiae length from the proximal growth plate to the tibiofibular junction - PGP TFJ. Results for graphs B-C presented as mean ± SEM, data analysed with two-way ANOVA, followed by Dunnett's post hoc. Results for graph D presented as median ± IQR, data analysed with multiple Mann–Whitney tests. **E** Methodology summary for the in vitro study of metatarsal elongation of E14.5 embryos (generated in BioRender). LAA-L ascorbic acid, βGP- β glycerophosphate. **F** Quantitation of metatarsal length as percentage increase. Results are presented as median ± IQR, data was analysed with Multiple Mann–Whitney tests. Data was obtained from N = 9, n = 34–40 *Clec14a⁺/⁺*

(WT) and N = 5, n = 13–16 *Clec14a⁻/⁻*, where N = individual biological replicates and n= total number of metatarsals analysed. **G** Representative μCT analysis images of 4- and 30-week-old *Clec14a⁺/⁺* (WT) and *Clec14a⁻/⁻* mouse tibias. **H** Micro-computed tomography (μCT) analysis of trabecular percent bone volume (BV/TV %), trabecular thickness (Tb.Th), trabecular number (Tb.N), trabecular separation (Tb.Sp). For BV/TV %, results are presented as median ± IQR, data was analysed with multiple Mann–Whitney tests. For the other parameters, results are presented as mean ± SEM, data was analysed with two-way ANOVA, followed by Dunnett's post hoc, compared to WT mice. **I** Representative μCT images of trabecular bone extent (blue). Note the increased trabecular bone extent in *Clec14a⁻/⁻* samples. **J** Quantification of trabecular bone extent as a percentage. For all graphs, statistical test results are shown as: *P < 0.05, **P < 0.01; ***P < 0.001; ****P < 0.0001. Sample size varies - each dot represents one biological repeat.

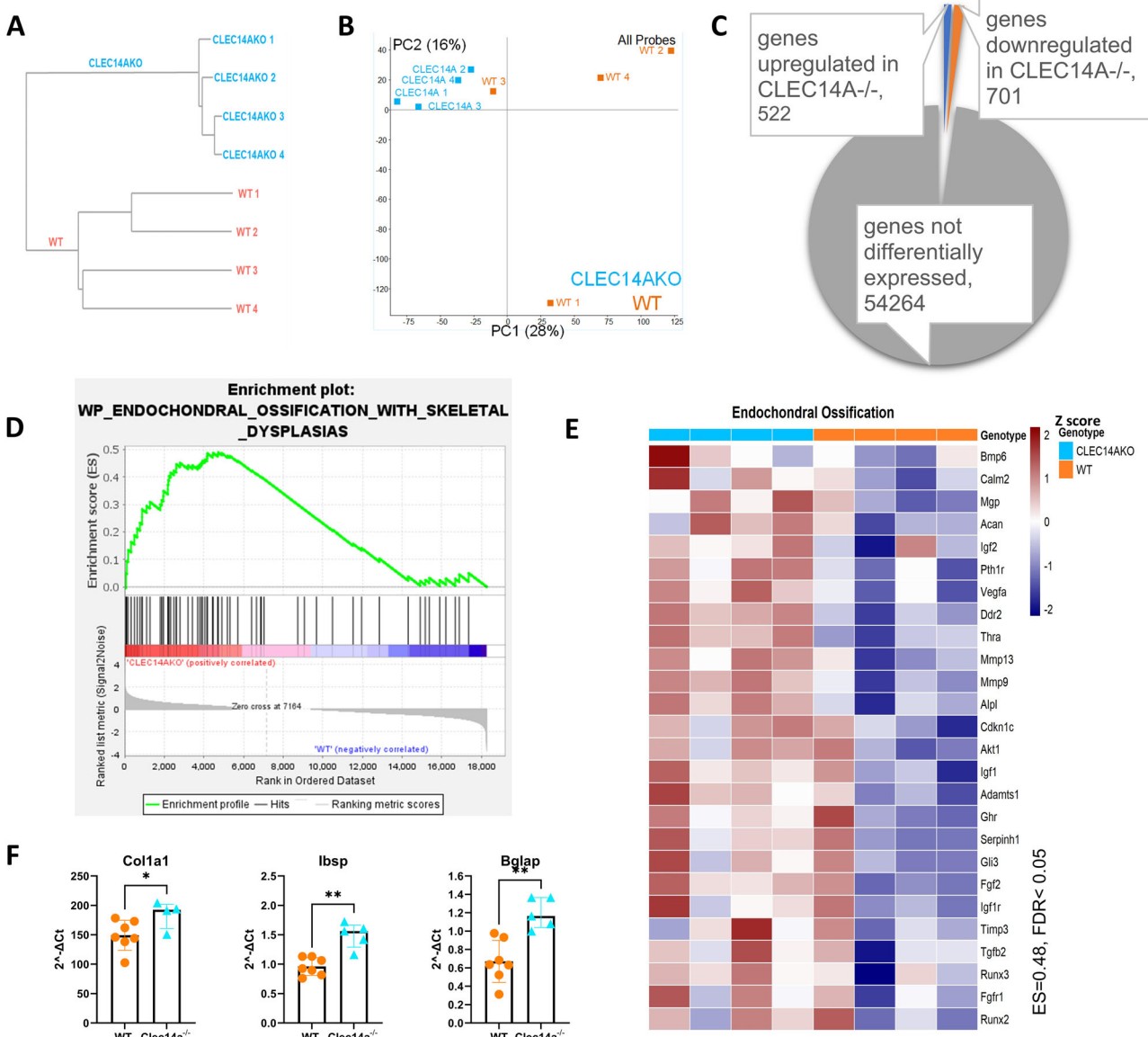

**Fig. 5 | RNA sequencing of molecular changes in *Clec14a*⁻/⁻ lysates.**
**A–E** Clustering and gene ontology analysis of *Clec14a*⁺/⁺ (WT) and *Clec14a*⁻/⁻ pup calvaria lysates. **A, B** Unsupervised hierarchical clustering, (dendrogram – A; principal component analysis – B) of log2 normalised counting reads over exons (RPM-reads per million) for all detected genes, N = 4 for both groups. *Clec14a*⁻/⁻ samples (blue), *Clec14a*⁺/⁺ samples (WT - orange). Each sample represents a different individual animal. C) Differentially expressed gene counts. Raw counts were analysed with DESeq2, significantly expressed genes were cut off at log2FC values > 0.5, adjusted *p* (padj) < 0.05, *p* value was corrected for multiple comparisons (Benjamini–Hochberg); Orange = genes upregulated in *Clec14a*⁻/⁻ samples; Blue = genes downregulated in *Clec14a*⁻/⁻ samples. **D** GSEA analysis enrichment score for pathways related to endochondral ossification. ES-enrichment score, FDR -false discovery rate. Gene expression is presented as a *Z* score (scale). **E** GSEA heatmap showing the list of leading-edge genes enriched at the top of the Wikipathways 'Endochondral Ossification' gene set. Heatmap presents upregulation of transcripts relevant to endochondral ossification in *Clec14a*⁻/⁻ (blue) samples compared to WT control (orange). Gene expression is presented as a *Z* score. Transcripts presented were identified as statistically significant by GSEA analysis, FDR (*q*-value) < 0.05, nominal *p* < 0.01. **F** Analysis of expression of genes linked to osteoblast maturation and osteogenic activity. P4 pup calvarial lysates were subjected to RT qPCR analysis to dissect expression patterns of genes known to regulate commitment to osteoblast lineage and osteoblast maturation, gene expression was normalised to Gapdh, shown as arbitrary units ($2^{\wedge -\Delta CT}$). Data is presented as median ± IQR, a Mann–Whitney test was used to analyse the results. Plots represent results from two independent experiments (n = 7 *Clec14a*⁺/⁺, n = 4–5 *Clec14a*⁻/⁻), data points in each graph are biological replicates.

mRNA expression of *Col1a1*, *Bglap* and *Ibsp* were confirmed in an independent qPCR experiment (Fig. 5F).

### EC interaction with osteoblasts supports osteoblast maturation
To test whether the cause of increased osteoblast maturation in *Clec14a*⁻/⁻ mice was due to signalling through its ligand Mmrn2, antibodies that bind Clec14a[22] were given for four weeks to 4-week-old wildtype mice. A measurable increase in trabecular number, trabecular surface, and trabecular density (BV/TV%) were detected following treatment with an anti-Clec14a

antibody that blocks the Clec14a-Mmrn2 interaction (C4) but not in response to an anti-Clec14a antibody that does not have Clec14a-Mmrn2 blocking activity[22] or vehicle control (Fig. 6A).

Expression of osteoblast maturation genes was examined in osteoblasts isolated from the calvaria of *Clec14a*⁺/⁺ and *Clec14a*⁻/⁻ neonatal P4 pups cultured in the absence of endothelial cells. Analysis of alkaline phosphatase (Alpl) transcript (Fig. 6B) and protein (Fig. 6C) levels were assessed by qPCR and ELISA respectively, over 21 days in vitro. *Alpl* mRNA levels increased in both *Clec14a*⁺/⁺ and *Clec14a*⁻/⁻-isolated osteoblasts in response to

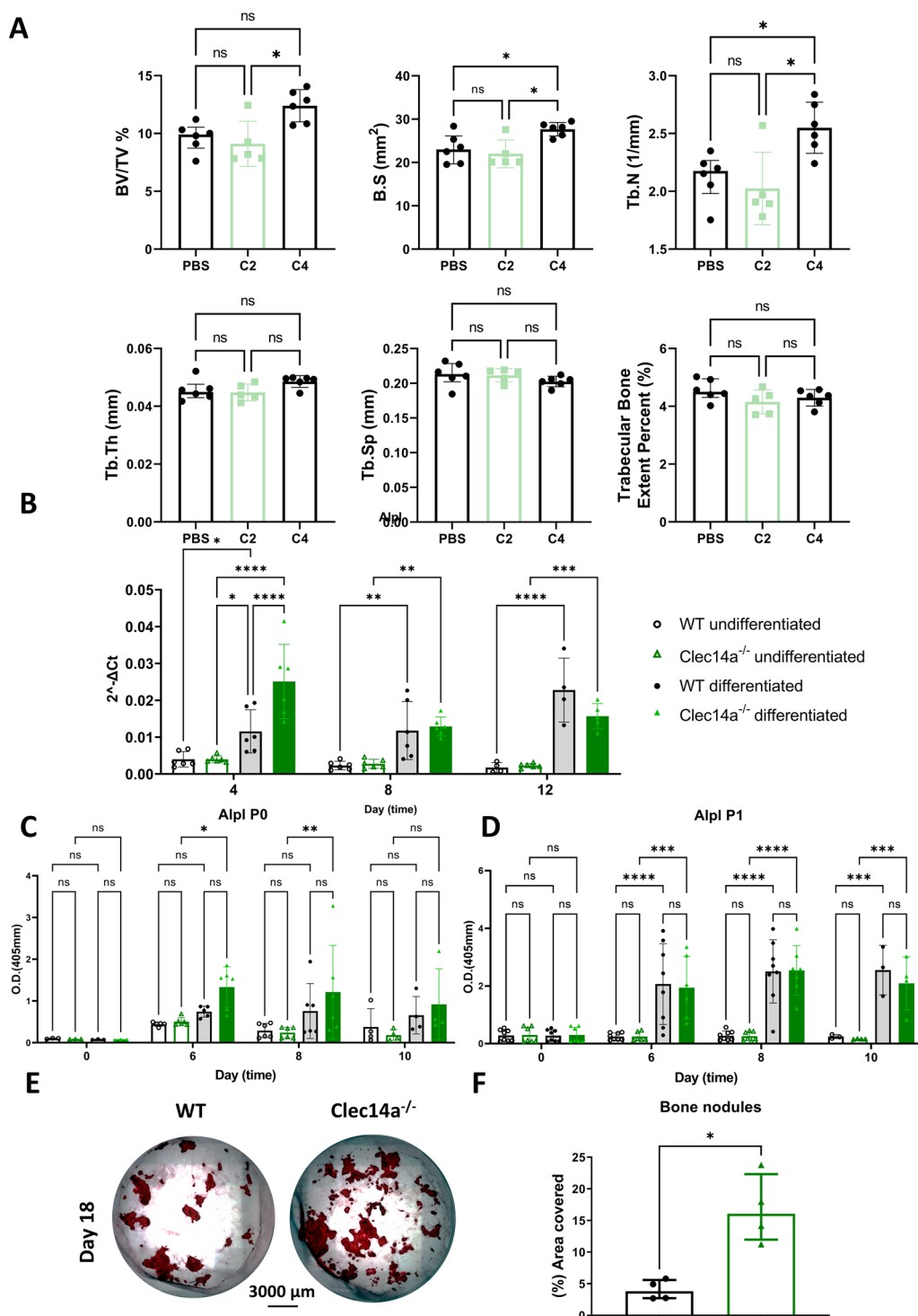

**Fig. 6 | Increased mineralisation activity of *Clec14a*$^{-/-}$-isolated osteoblasts in vitro. A** Bone morphometric parameters in the mouse tibia of C2- and C4-treated mice. Note, C4 blocks and C2 does not block the Clec14a-Mmrn2 interaction, as previously reported[22]. Micro-computed tomography (μCT) analysis of trabecular percent bone volume (BV/TV%), bone surface area (B.S), trabecular number (Tb.N), trabecular thickness (Tb.Th), trabecular separation (Tb.Sp) and trabecular extent. For BV/TV %, results are presented as median ± IQR, data was analysed with multiple Mann–Whitney tests. For the other parameters, results are presented as mean ± SEM, data was analysed with two-way ANOVA, followed by Dunnett's post hoc, *$p < 0.05$. **B** *Alpl* mRNA expression was analysed by RT qPCR in calvarial osteoblasts. Results were analysed with a two-way ANOVA followed by Dunnett's correction for multiple testing, data is presented as mean ± SD. **C, D** Murine cal- varial osteoblasts were isolated from *Clec14a*$^{+/+}$ (WT) and *Clec14a*$^{-/-}$ neonatal pups

at postnatal day 4 and osteogenesis was induced with osteoinduction media in (**C**) freshly isolated cells prior to passage (P0) or (**D**) in cultured cells after passage (P1). Alpl activity was measured by ELISA after 0, 6, 8 and 10 days in culture, in *Clec14a*$^{+/+}$ and *Clec14a*$^{-/-}$ osteoblasts that were incubated with either expansion media (undifferentiated – see key on graph) or osteoinduction media (differentiated – see key on graph). Data is presented as optical density (O.D.). Results were analysed with a two-way ANOVA followed by Dunnett's post hoc, data is presented as mean ± SD. *$p < 0.05$, **$p < 0.01$, ***$p < 0.001$. **E** Representative images of alizarin red stained osteoblast-deposited calcium following 18 days in culture. **F** Alizarin red coverage as percentage. Results were analysed with a Mann–Whitney test, data is presented as median ± IQR, *$p < 0.05$. For all graphs sample size is represented by each individual dot in the graphs.

maturation stimuli (osteoinduction media), however *Clec14a*$^{-/-}$-isolated osteoblasts reached peak *Alpl* expression at day 4, whilst peak expression in *Clec14a*$^{+/+}$ osteoblasts was observed 8 days later, at day 12 (Fig. 6B,C). Similarly, freshly isolated passage (P0) *Clec14a*$^{-/-}$-isolated osteoblasts had increased Alpl activity compared to *Clec14a*$^{+/+}$ controls in response to osteoinduction media at days 6 and 8 (Fig. 6C). Considering the lack of expression of Clec14a on osteoblasts, we theorised that the accelerated ossification of *Clec14a*$^{-/-}$-isolated calvarial osteoblasts was due to stimulation by endothelial cells in vivo prior to osteoblast isolation. To test this hypothesis, Alpl protein levels were examined in passaged (P1) *Clec14a*$^{+/+}$ and *Clec14a*$^{-/-}$-isolated osteoblasts, demonstrating complete abrogation of the phenotype (Fig. 6D). To establish if the increases in Alpl activity detected in freshly isolated *Clec14a*$^{-/-}$-isolated osteoblasts corresponded with increased mineralisation, alizarin red analysis (percentage area covered) was performed, which demonstrated a significant increase in *Clec14a*$^{-/-}$ samples compared to *Clec14a*$^{+/+}$ at day 18 (Fig. 6E, F).

## Discussion
Here, we show that expression of Clec14a and Mmrn2 is restricted to endothelial cells in the bone stroma, and that both genes are highly differentially regulated, with markedly increased expression in type-H compared to type-L endothelium. Previous studies have identified a link between irregular or disturbed blood flow in tumours and increased expression of Clec14a[37]. Given the increased blood velocity and shear stress rates previously reported in type-H vessels[8], the higher transcriptional activity of genes sensitive to shear stress (such as Clec14a) is likely triggered in response to the blood flow dynamics in these vessels, especially in the blunt H-type capillary bud area, which has disturbed blood flow[5,8,36,38].

Previous in vitro investigations have shown that genetic deletion of *Clec14a* alters sprouting angiogenesis by diminishing EC sprout formation capacity and downregulating EC tube formation[22,39]. Genetic deletion of *Clec14a* in vivo also resulted in compromised angiogenesis responses where vascular invasion into subcutaneously implanted sponges was impaired and accompanied by a diminished vascular density[40]. Against this background, it was vital to assess the impact of germline deletion of *Clec14a* on the vascularisation of the bone. To achieve this, we employed 3D high-resolution immunofluorescent imaging[41] of *Clec14a*$^{-/-}$ and *Clec14a*$^{+/+}$ bone sections to enable detailed morphological characterisation of the vascular network. In wildtype (*Clec14a*$^{+/+}$) mice, the advancement of the vascular front at the growth plate-metaphysis interface appears uniform and straight whilst the leading vascular front in sections from *Clec14a*$^{-/-}$ mice (either neonatal or at 4 weeks old) was less uniform. We also noted that tip cell filopodia formation at the vascular front was reduced in *Clec14a*$^{-/-}$. These findings indicate dysregulation of tip cell formation and EC migration, in keeping with the role previously attributed to CLEC14A in vascular development - genetic deletion of *Clec14a* results in reduced EC filopodia formation and migration[39].

In addition, blunting of type-H vessel buds at the growth plate was observed in 4-week-old but not younger P4 *Clec14a*$^{-/-}$ mice. Previous reports have suggested that blunting of the type-H vessel buds indicates maturation of the vascular compartment in the metaphysis as bone growth begins to decelerate[5,8–10]. It was not possible to categorically confirm whether the phenotype observed in *Clec14a*$^{-/-}$ mice is predominantly caused by acceleration of type-H vessel maturation or because of inhibition of vascular elongation. Shortening of the type-H vascular front at 4 weeks but not P4 in *Clec14a*$^{-/-}$ and the decreased percentage of Cd31$^{high}$ Emcn$^{high}$ EC levels in 2-week-old mice (the time point where peak type-H / type-L EC ratio occurs), supports the hypothesis that accelerated maturation of the compartment is occurring. To confirm whether vascular elongation at the chondro-osseous interface is also inhibited and responsible for the bone phenotype observed in the *Clec14a*$^{-/-}$ mouse will require further experimentation with labelled transgenic mice. Of note, given the reduced expression of Clec14a protein on type-L vasculature, the morphology and organisation of the type-L vasculature was not studied in detail in this manuscript.

Longitudinal bone growth is initiated through the angiogenic expansion of the blood vessel front, triggering osteoblast co-migration and subsequent osteoblast-mediated synthesis of bone extracellular matrix[4,5,7–9,11,36,38,42,43]. To assess the role of Clec14a in this process, we investigated osteoblast migration to, and maturation at, the metaphysis by characterising the patterning of immature osteoblasts in the neonatal (P4) and juvenile (4-week) murine tibia of *Clec14a*$^{+/+}$ and *Clec14a*$^{-/-}$ mice. Imaging analysis identified an increase in the region covered by osteoblasts with an extension towards the tibial diaphysis. These changes in osteoblast localisation and Col1a1 secretion are likely to be linked to a perturbed interaction between EC and osteoblasts, given the well-known ability of vascular bone capillaries to regulate patterning of Col1a1 deposition and mineralisation of the bone matrix by acting as guiding templates[44]. This was further evidenced by antibody-mediated inhibition of the Clec14a-Mmrn2 interaction, which recapitulated the *Clec14a* genetic deletion phenotype. Given that genetic deletion of Cd248 (a cell surface protein highly expressed by osteoblasts) also results in a similar high bone mass phenotype[30] and that the CLEC14A-MMRN2-CD248 interactome is known to occur in other tissues[22] we postulate that the interaction maintains immature osteoblasts in close proximity to the type-H endothelium and delays their differentiation to bone-forming mature osteoblasts to regulate ossification.

Whilst detailed characterisation of osteoblast maturation state is challenging in ex vivo tissue sections, in vitro culture of calvarial osteoblasts freshly isolated from *Clec14a*$^{-/-}$ mouse newborn pups also demonstrated increased propensity to maturation as evidenced by increased alkaline phosphatase at the mRNA and protein levels along with increased bone nodule formation. This phenotype was lost following passage of the primary cells. The finding that freshly isolated osteoblasts from *Clec14a*$^{-/-}$ mice more readily differentiate and produce ossified matrix when exposed to differentiation media indicates that priming has occurred in vivo in the context of *Clec14a*$^{-/-}$ endothelial cells. Evidence of this priming at the gene expression level was identified by bulk RNA sequencing and RT qPCR analysis of *Clec14a*$^{-/-}$ calvarial lysates, which showed enrichment of osteoblast maturation markers compared to *Clec14a*$^{+/+}$ samples.

The effect of *Clec14a* genetic deletion was evident in accelerated trabecular bone formation and was closely linked to type-H vessel migration in the bone anlagen. We saw no overt effects on cortical bone morphometric parameters. Similarly, the effect of Clec14a genetic deletion on bone density was transient, resolving as bones reached skeletal maturity, in keeping with the transient nature of type-H vasculature in the bone metaphysis.

The role of type-H EC in fracture healing[5,12,45,46], and recent interest in harnessing them to support osteogenesis in osteoporosis[17,47–50] is evidence of the potential of this cell type as a therapeutic target. To realise this potential, identification of druggable targets that control the interaction of type-H EC with osteoblasts is crucial. Here we identify Clec14a as a key endothelial cell-specific target protein, whose inactivation, increases bone formation. Identification of entities that can modulate the activity of this protein will facilitate the dissection of the signalling pathways through which Clec14a exhibits its effects and may hold potential as novel bone anabolic therapeutics.

## Methods
### In vivo models
*Clec14a*$^{-/-}$ mice[40] were maintained at the Biomedical Services Unit, University of Birmingham. The mice were backcrossed to C57BL/6J mice and refreshed regularly. Genotype and genetic drift were assessed by SNP genetic monitoring (Transnetyx Inc, Cordova). C57BL/6 J control (WT-*Clec14a*$^{+/+}$) mice for experiments and to refresh the *Clec14a*$^{-/-}$ colony were obtained as required from Charles River, UK. Mice had *ad libitum* access to food and water and were maintained at 21 °C with 12-h light/dark cycles. All in vivo experiments were performed under Project Licence PFBB3DA4C in accordance with the UK Animal (Scientific Procedures) Act 1986 and approved by the local ethics committee (BERSC: Birmingham Ethical Review Subcommittee). Mice were humanely sacrificed by cervical dislocation. We have complied with all relevant ethical regulations for animal

use. Male mice were used for all experiments shown, at the ages indicated in each figure. Experiments with female mice were also performed and these data are included in the Supplementary Figures.

In house monoclonal antibodies, hereafter referred to as C2 and C4 were generated by Noy et al. and used to either bind Clec14a without blocking its interaction with MMRN2 (C2) or to bind Clec14a and block its interaction with MMRN2 (C4)[40]. 100 µg of C2 or C4 antibody or sterile PBS (vehicle control) were intraperitoneally administered to 4-week-old C57BL6/J male mice twice a week for 4 weeks.

## Analysis of gene expression by RNA sequencing
Total RNA was isolated from dissected calvaria of WT (Clec14a[+/+]) and Clec14a[−/−] pups at postnatal day 4 (P4) as previously described[51]. Note, collagenase D (Sigma Aldrich Roche, UK, cat no: COLLD-RO) was used instead of collagenase I. Total RNA was extracted from isolated osteoblasts using the Qiagen RNeasy Micro Extraction Kit (Qiagen, UK, cat no: 74004) and RNA integrity and quantity analysed with High Sensitivity RNA ScreenTape® (Agilent, California) and purity determined using the Qubit High Sensitivity RNA assay kit (ThermoFisher Scientific, UK, cat no: Q10210). RNA quantity per sample was between 102 and 234 ng µl$^{-1}$ and all RNA integrity number (RIN) scores were >9.

50 ng RNA per sample was used to generate libraries using the Lexogen QuantSeq 3′ mRNA-Seq Library Prep Kit FWD for Illumina (Lexogen, US) and their quality assessed as above. Equal amounts (4 nM per sample) were sequenced: v2.5 150 cycles mid output (75 single read) flowcell, NextSeq 500 (Illumina, California). The data has been uploaded to the SRA database with accession number PRJNA1027335.

Separately, analysis of a publicly available datasets was performed: 1. Bulk RNA sequencingAccessed on January 28th, 2020, via the ArrayExpress database http://www.ebi.ac.uk/arrayexpress/), accession number E-MTAB-4066, was performed[9]. The dataset was generated by Langen and colleagues as follows: RNA sequencing (RNASeq) was performed on samples pooled from 20 C57BL/6J mice at postnatal day 6. Endothelial cells were sorted by flow cytometry into type H and L vessels, based on the level of Endomucin and Cd31 expression. Sequencing libraries were prepared using 100 ng of RNA.

Raw sequences were quality control checked using FastQ. Raw reads were then trimmed to remove low quality reads and adaptors using Cutadapt (Bioconductor: https://bioconductor.org), then mapped to the mouse genome assembly GRCm38 (mm 10). Quantitation of mRNA transcripts was achieved using SeqMonk software (Babraham Institute, Cambridge, UK): All empty transcripts (value = 0) or those corresponding to pseudo-genes or non-coding RNA were excluded from subsequent analysis, and transcript isoforms were merged as a single measurement for each gene. Data was corrected for potential DNA contamination by evaluating the read density in intergenic regions as a basis for estimating DNA contamination. Using this estimate, individual corrections were applied to each transcript to increase the count accuracy. For quality control purposes all samples were verified for the percentage of genes falling into exons, ribosomal or mito-chondrial RNA, based on raw counts. Pearson correlation was employed to calculate a distance matrix between the 6 data stores (3 samples for type-H ECs and 3 samples for type-L EC), this analysis was then used to construct a neighbour joining tree from the data matrix.

Quantitation of differential gene expression was performed in Seq-Monk with a negative binomial generalised linear model (DESeq2) or by testing for variance of gene expression intensity difference. Similar results were obtained for quantitation of reads per gene with both methods, therefore DESeq2 data are reported here. Differentially expressed genes were reported for an adj.$p < 0.05$ and a log2 fold change >0.5 and <−0.5, for a more stringent cut off point of a log2 fold change of >1.5 and <−1.5 and an adj.$p < 0.05$. 434 genes were differentially expressed, 46 were upregulated while 388 transcripts were downregulated in Clec14a[−/−] calvaria lysates. All reported values for DESeq2 analysis were adjusted for multiple testing with the Benjamini–Hochberg method to control for false discovery rate (FDR). Heatmaps of differentially expressed genes were produced using R (The R

Foundation for Statistical Computing, Vienna, Austria). Gene set enrich-ment analysis was accomplished using the GSEA software (MSigDBv6, Broad Institute)[52], using gene lists with log2 normalised gene sets, gene lists were probed against the Hallmark, Kyoto Encyclopaedia of Genes and Genomes (KEGG), and Reactome gene sets from the GSEA signatures database collection.

For single-cell (sc) sequencing dataset analysis, genes, barcodes and matrix files were downloaded from the Gene Expression Omnibus website for publicly available dataset GSE128423[51], accessed on March 1st 2022 (https://www.ncbi.nlm.nih.gov/geo/query/acc.cgi?acc=GSE128423). Data was mined to identify Clec14a and Mmrn2 mRNA expression levels on cells isolated from the bones of 8-week-old C57BL/6J mice[51]. To generate this dataset, individual objects were loaded and merged into a single data matrix. All cells that had more than 200 and fewer than 6000 transcripts and lower than 5% mitochondrial gene transcripts were sub-set and analysed further. All reads were log normalised at a scale factor of 10$^3$ and variable features were identified to produce a set of genes that was most variable across the single-cell data set. Unsupervised dimensional reduction (tSNE) was con-ducted on the single-cell object using the variable features identified in the previous step to ensure identification of the primary structures in the dataset. Neighbouring cells and cell clusters were identified with the aid of the FindNeighbors and FindClusters commands.

## Gene expression analysis from murine postnatal calvarial osteoblasts
Total RNA was extracted from calvarial osteoblasts with the RNeasy Micro Kit (as above) immediately after cell isolation (as above) or at the passage indicated. Residual genomic RNA was removed with DNase I digestion and the quantity and purity of total RNA assessed using a Nanodrop 2000 analyser (ThermoFisher, UK). 500 ng RNA per sample (equalised in RNAse free water) was added to a MicroAmp™ Optical 96-Well Reaction Plate (ThermoFisher, UK, cat no: N8010560) at a 1:1 ratio of RNA to cDNA master mix (High-Capacity cDNA Reverse Tran-scription Kit, ThermoFisher, UK, cat no: 4368814). RNA was denatured for 5 mins at 65 °C and reverse transcribed for 1 h at 50 °C. Gene expression and reference gene Gapdh levels were detected using Applied Biosystems® TaqMan® Gene Expression Master Mix and FAM-conjugated Taqman probes (Supplementary Table 1) in a Roche LC480 real-time thermal cycler in a reaction volume of 5 µL consisting of a mix of: 2.52 µl Applied Biosystems® TaqMan® Gene Expression Master Mix, 0.28 µl relevant Taqman primer, and 2.3 µl cDNA per sample. Amplifi-cation conditions: 95 °C for 2 min followed by 40 cycles of 95 °C for 15 s and 60 °C for 1 min.

## Flow cytometry
Mouse tibiae and femora were dissected from 28-day-old mice and all adjacent muscle and connective tissue was removed. The epiphyses were discarded, and the remaining metaphysis and diaphysis regions were cru-shed in ice-cold PBS (Oxoid, UK, cat no: BR0014) and digested in 1 mg ml$^{-1}$ type A collagenase (Roche Diagnostics, Germany, cat no: 10103586001) in PBS for 20 min at 37.5 °C with constant agitation. The resulting cell sus-pension was filtered (100 µm pore size) and washed with MACS buffer (10% FBS, Biosera FB1001; 0.002 M EDTA, Merck 324506-100 ML; 0.1% sodium azide (NaAz), Sigma-Aldrich, cat no: S2002) and subjected to red blood cell lysis (Roche, UK, cat no: 454643000).

Following blocking with 1:100 anti-CD16/32 antibody (eBioscience, 14-0161-82) in MACS buffer (20 min, on ice), $1 \times 10^7$ cells were immu-nostained for 45 min on ice with the following antibodies (all diluted 1:200 in MACS buffer): Anti-Emcn conjugated to FITC, Santa Cruz, sc-65495 FITC; anti-Cd31 conjugated to PE, Biolegend, 102408; anti-Cd45 con-jugated to PerCP-Cy5.5, Biolegend, 103130. Anti-Ter119 antibody con-jugated to PerCP-Cy5.5 (dump channel), Biolegend 116226, was used at a dilution of 1:100. Cell viability was assessed using 1:500 Zombie NIR (Biolegend, 423105) and all data were acquired on a Fortessa flow cytometer (BD Biosciences, US) and analysed using Flow Jo (FlowJo LLC, US).

https://doi.org/10.1038/s42003-024-06971-3                                                                                    **Article**

The percentage of type-H EC was quantified as live cells/Cd45$^-$/Ter119$^-$/Cd31$^{high}$/Emcn$^{high}$, whilst the fraction of cells corresponding to L-type EC was quantified as live cells/Cd45$^-$/Ter119$^-$/Cd31$^{low}$/Emcn$^{low}$. This strategy was employed to distinguish type-H EC from Emcn$^-$ arterial and Emcn$^{low}$ L-type EC sinusoidal and venous cells. The gating strategy is shown in Supplementary Fig. 10.

### 3D immunofluorescent staining

Mouse tibiae were prepared as described in Kusumbe et al. 2015[41]. Briefly: Longitudinal 100 μm thick sections were cut (Leica CM1950 cryostat – chamber temperature at −23 °C, specimen head temperature at −21 °C), air-dried overnight and stored at −20 °C. Sections were permeabilised in 0.3% Triton X-100 (Sigma-Aldrich, UK, X100-500ML), blocked for 30 min in 10% horse serum (Sigma-Aldrich, UK, H0146) diluted in PBS and then incubated with primary antibodies (1:100 anti-Emcn, Santa Cruz, US, cat no: sc-65495, 1:100 anti-CD31, BD Biosciences, cat no: 553370, 1:500 anti-Osx, Abcam, UK, cat no: ab227820, 1:100 anti-Col1a1, EMD Millipore Corporation, US, cat no: AB749P, anti-Ki67 1:100, eBioscience, 14-5698-82) for 1 h at room temperature, washed with PBS and incubated with Alexa Fluor-conjugated secondary antibodies (AF546, Invitrogen, A11081; and 1:100 anti-rabbit AF647, Jackson Immuno Research, US, 711-606-152, 1:400 Hoechst diluted in MACS). Before imaging, sections were mounted with Prolong Diamond Antifade Mountant (Thermo Fisher Scientific, UK, cat no: P36961). Apoptosis was detected using a TUNEL assay, Invitrogen, cat no: C10247, as per the manufacturer instructions.

Imaging was performed on an LSM880 confocal microscope. Following acquisition, images were Airyscan processed with ZenBlack and analysed with Image J. Total number of osteoblasts was quantified in ImageJ using the "Analyse Particles" function following image thresholding and cell segmentation of the image mask. The following settings were utilised for the "Analyze particles" plugin: Size 0.02-Infinity, m2, Circularity 0.1–1.0, transformed with a Gaussian blur filter of a sigma radius of 0.5, automated thresholding was applied utilising the "Li" method. Cells were segmented using the "Watershed" function. Osteoblast nuclear circularity was calculated in Image J with the "Analyse Particles" function as: $4p\pi(Area)/(Perimeter^2)$. Osteoblasts proximal to the periosteum were excluded from the analysis for all samples. Further details of the imaging analysis strategies employed are detailed in Supplementary Fig. 4.

For the in vivo labelling of Clec14a in the mouse bone, an in house monoclonal anti-Clec14a antibody (clone C2)[40] was labelled with biotin (Sigma-Aldrich, UK, MXBIOS50-1KT). 100 μg biotin-conjugated antibody diluted in sterile PBS was injected intraperitoneally and hind legs were collected 24 h post injection and tissue was prepared as described above. Sections were probed with streptavidin APC (ThermoFisher Scientific, UK, cat no: SA1005).

Illustrations were produced in Inkscape, Inkscape Project, 2020, available at: https://inkscape.org or Adobe Illustrator, available at: https://adobe.com/products/illustrator or BioRender, available at: https://www.biorender.com/.

### Bone length and structural analysis

Callipers (Digi-Key, UK, cat no: 3C301-NB) were used to measure the length of tibiae by measuring the distance between the proximal and distal ends of the epiphysis. To ensure high reproducibility, callipers were placed between the medial and lateral condyle in the intercondylar fossa at the distal end of the femur and in the trochanteric fossa for the proximal end.

Metatarsals were isolated from E14.5 $Clec14a^{+/+}$ and $Clec14a^{-/-}$ embryos under sterile conditions[53]. Isolated metatarsals were cultured in 24 well plates in 300 μl osteoinductive media (αMEM media (Sigma) supplemented with 10% FBS, 2 mM L-Glutamine, 100 U/ml Penicillin, 100 μg ml$^{-1}$ Streptomycin, 50 μg ml$^{-1}$ L-ascorbic acid (Sigma, UK, cat no: A5960) and 10 mM β-glycerophosphate (Sigma, UK, cat no: G9422) for 7 days at 37 °C and 5% CO$_2$. Media was refreshed on day 5. Metatarsal

length was measured at days 1, 3, 5 and 7 through imaging on a Zeiss Primostar 3 widefield microscope and images analysed in Image J using the straight-line tool.

Bone structural analysis was performed using micro-computed tomography (μCT) with a Skyscan 1172 Micro-CT scanner (Bruker, Belgium)[30]. Image acquisition was performed at voltage of 60 kV and current 167 μA. Multiple projections were acquired at 580 ms exposure over 360° rotation with a rotation step size of 0.45°, frame average of 4 and random movement of 0. The projections were utilised to generate 3D reconstructions using NRecon (version 1.4.4, SkyScan, Belgium) with beam hardening parameters of 20% and ring artefacts correction of 4. The resulting 3D dataset had 1200 slices with voxel size of 14.5 μm. Bone structures were analysed digitally using CTAn (version 1.7.0.2, SkyScan, Bruker). When calculating trabecular bone parameters, a standardised region of the tibial proximal metaphysis was assessed, the analysed area extended 1370 μm from the lowest slice of the proximal growth plate towards the diaphysis. The extent of trabecular bone from the metaphysis to the diaphysis was measured in a cranial-caudal direction. Cortical morphology was evaluated from 100 slices morphed from a 1370 μm region of the midshaft, whose superior point was identified as 250 slices from the lowest slice in the proximal growth plate. All analysis was performed blinded.

### Osteoblast maturation assays

Osteoblast lineage-enriched cells were isolated from $Clec14a^{+/+}$ and $Clec14a^{-/-}$ postnatal day 4 pup calvaria under sterile conditions as described above and previously[30,54]. Cells were seeded into a 48 well plate at a density of $8 \times 10^3$ per well and grown to confluence. Subsequently, cells were incubated with 'osteoinduction' media: αMEM media (Sigma) supplemented with 10% FBS, 2 mM L-Glutamine, 100U ml$^{-1}$ Penicillin, 100 μg ml$^{-1}$ Streptomycin, 50 μg ml$^{-1}$ L-ascorbic acid (Sigma, UK, cat no: A5960) and 10 mM β-glycerophosphate (Sigma, UK, cat no: G9422) for 21 days. As a control, and for expansion/passaging of cells, osteoblasts were also separately cultured in 'expansion' media lacking β-glycerophosphate and L-ascorbic acid.

Alkaline phosphatase activity was measured in cell lysates by ELISA (Sigma, UK, cat no: P7998-100). Cell lysates were mixed with alkaline phosphatase yellow stain at a ratio of 1:5 in a final volume of 100 μl of solution then incubated with constant gentle agitation for 30 min at 37 °C. Alpl activity was measured in osteoblast cell lysates with the use of a buffered alkaline phosphate substrate containing p-nitrophenyl phosphate (pNPP), read at 405 nm on the BioTek® synergy HT plate reader and analysed using BioTek® Gen5 plate reading software.

Bone nodule formation was studied with the aid of alizarin red staining on formalin-fixed (10% formaldehyde, for 15 min at RT) cultures: 40 mM alizarin red stain (ScienCellTM, UK, cat no: 8678) under constant gentle agitation (~90 rpm) for 30 min, at RT, in the dark. Images were acquired at 4X magnification using the Cytation 5 microscope (Biotek/Agilent, United States) and analysed in ImageJ (NIH, United States) by mapping a colour threshold over the acquired images, to give percent of area covered by alizarin red staining. The same protocol was also utilised when staining mineralisation in the P4 C57Bl/6J pups calvaria.

### Statistics and reproducibility

Parametric tests were employed if data was continuous, normally distributed and presented equal variance. If one of these requirements was not satisfied, non-parametric testing was employed. For parametric data, a Student's t-test or a parametric one-way ANOVA were used to compare two groups, or to compare three groups or more, respectively. For non-parametric data, a Mann–Whitney U test or a Kruskal Wallis test were employed to compare two groups, or to compare three groups or more, respectively. When two dependent variables were compared, a two-way ANOVA was employed for parametric data, or alternatively, multiple Mann–Whitney U tests were employed to compare two groups at a time if data was non-parametric. The sample size is stated for each experiment in the figure legends. For all in vivo experiments power calculations were used

to calculate the required sample size. This was accomplished with the aid of G Power v3.1 (Allgemeine Psychologie und Arbeitspsychologie, Dusseldorf, Germany).

## Reporting summary

Further information on research design is available in the Nature Portfolio Reporting Summary linked to this article.

## Data availability

The bulk RNA sequencing data generated from wildtype and *Clec14a⁻ᐟ⁻* calvarial lysates has been uploaded to the SRA database with BioProject accession number PRJNA1027335 and is freely available. All other data is provided within the figures and Supplementary Figs. and the source data has been uploaded as Supplementary Data 1.

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

## Acknowledgements
This work would not have been possible without the support of Delia Margaret Goodall, Manager Monoclonal Antibody Production Unit, University of Birmingham. A.J.N. was supported by a Career Development Fellowship from Versus Arthritis #21743. G.N. and J.L. were supported by PhD studentships from the Centre for Musculoskeletal Ageing Research (CMAR) funded by the MRC and Versus Arthritis. This paper represents an independent research part-funded by the MRC-Versus Arthritis Centre for Musculoskeletal Ageing Research. The views expressed are those of the authors and not necessarily those of the MRC or Versus Arthritis. This work was sponsored by grant number MR/P021220/1. J.E.M. was supported by a Medical Research Council (MRC) Industry Case Studentship (MR/P016154/1). We acknowledge the advice received from Anjali Kusumbe, University of Oxford, who recommended type A collagenase for flow cytometry cell preparation.

## Author contributions
A.J.N. and R.B. conceived the experiments and supervised the project. G.N. performed most of the experimental work with practical and analytical support from J.L., J.D.T., J.E.M., I.D., M.F., G.P., J.W., M.A.S., K.A.K. and J.B. Resource/material access and supervision was provided by H.M.M., I.B., V.H., S.W.J. and C.D.B.; G.N. and A.J.N. wrote the manuscript and analysed the results, with additional input from C.D.B. and R.B. All authors have been involved in the writing and editing of the manuscript.

## Competing interests
The authors declare no competing interests.
