## [Peer Review File · Communications Biology]

Reviewers' comments:

Reviewer #1 (Remarks to the Author):

This research was novel and original, and I think it will interest others in the field.

This study suggests that Clec14a is a key endothelial-cell-specific target protein, and the effect of Clec14a genetic deletion was evident in accelerated trabecular bone formation and was closely linked to type-H vessel migration in the bone anlagen.

But there are some confused results of this work.

1) According to the previous study, in vitro investigations have shown that genetic deletion of Clec14a alters sprouting angiogenesis by diminishing EC sprout formation capacity and downregulating EC tube formation (Clec14a genetically interacts with Etv2 and Vegf signaling during vasculogenesis and angiogenesis in zebrafish. *BMC Dev Biol.* 2019;19(1):6.), and genetic deletion of Clec14a in vivo, also resulted in compromised angiogenesis responses where vascular invasion into subcutaneously implanted sponges was impaired and accompanied by a diminished vascular density (Blocking CLEC14A-MMRN2 binding inhibits sprouting angiogenesis and tumor growth. *Oncogene*, 2015, 19;34(47):5821-31.), those both proved the importance of Clec14a for type H blood vessels. In this study, Clec14a^{-/-} in vivo results in a decline of the type-H vascular front, but at the same time, accelerated trabecular bone formation. Please explain how type H vessels are linked to osteogenesis and the molecular mechanism.

2) The evidence from this study confirms the relationship between Clec14a and type H vessels and osteogenesis. Still, there is insufficient evidence to show the signaling pathway through which Clec14a affects angiogenesis and osteogenesis, which needs to be further confirmed by additional data.

3) I don't think this paper will influence the thinking in the field.

Reviewer #2 (Remarks to the Author):

In their manuscript with the title "Type-H endothelial cell protein Clec14a orchestrates osteoblast activity during trabecular bone formation and patterning", Neag et al. report that C-type lectin domain containing 14A (Clec14A) is an important regulator of postnatal vessel growth in bone. Clec14a knockout mice display a reduction of bone vessels and, in particular, of type H capillaries, a subset of vessels that is associated with perivascular osteoprogenitors and is involved in the regulation of osteogenesis. The authors also show that skeletal development and osteoblast

maturation are accelerated in postnatal Clec14a mutants. Further data indicates that the secreted protein Multimerin 2 (Mmrn2), which is highly expressed by type H endothelial cells, is involved in the communication with osteoblast lineage cells.

Overall, the presented findings are novel and interesting, but revisions are necessary to clarify several important issues.

Specific questions:

It is argued that Clec14a and Mmrn2 protein expression is associated with type H vessels in postnatal bone. The relevant stainings in Fig. 1, however, lack any endothelial marker. Thus, this claim seems not sufficiently supported by the data. Likewise, it would be helpful to include a specific marker in support of the arterial localization of Mmrn2.

It remains unclear how Clec14a is actually regulating angiogenesis in bone and some explanation is needed. Are EC proliferation or survival altered in the knockout mice? Do type H cells lose their characteristic features?

It is stated that type L sinusoidal vessels are not altered, but the representative images are not sufficiently good to support this claim. In particular, the diameter or branching pattern of sinusoidal vessels appears different in the images showing mutant samples.

It is argued that loss of Clec14a changes vessel buds at the chondro-osseous interface. The images provided support some sort of change, but it would be very helpful to combine the vessel staining with a chondrocyte marker. Ultrastructural data would be useful or, if this cannot be obtained, higher magnification images.

The authors propose that osteoprogenitor maturation is accelerated in Clec14a mutants, but Osx is shown as the only marker. It would be important to look at Runx2+ cells.

The calvaria data is interesting, but it is much less clear whether (and how) alterations in the vasculature and bone-forming cells cause this phenotype. This is a weakness and therefore this data is not adding much insight.

Clec14a is presented as a negative regulator of osteogenic differentiation/maturation, but this is not entirely clear on the basis of the data provided. Can the authors check whether the association of osteoprogenitors and vessels is altered in Clec14a mutants, which might lead to partially uncoupled angiogenesis and osteogenesis. The data in Suppl. Fig. 5 is noted, but higher magnification images and quantification are needed.

Minor comments:

The Discussion recapitulates a lot of aspects from the Results and could be trimmed or improved. It is important to discuss the transient nature of the vessel defects in Clec14a mutants. Are there

redundant gene products that get upregulated in adolescence or is Clec14a expression or function confined to growing vessels?

Fig. 3A: *Osx*⁺ cells in the diaphysis are presumably associated with the endosteum. Please clarify.

While it is not a focus of this study, weaker DAPI signal (see Suppl. Fig. 5) indicates potential changes in the *Clec14a* mutant bone marrow. Can you please comment?

Suppl. Fig. 5: It is argued that “*Osx*⁺ cells were also detected in the growth plate”. Presumably, this is *Osx* expression by chondrocytes? Please clarify.

Fig. 4G: Are control and mutant calvaria shown in the same anterior-posterior orientation?

The word “culture” is missing in line 4 of the paragraph beginning with “In vitro culture of calvarial...”.

Response to reviewers

Please find below our response to reviewers. We are grateful for the thorough review of our manuscript. We include below the entire review text and have addressed each point in turn. Manuscript changes and additions that we have made in response to each comment are detailed and linked to the figure part or manuscript part, as appropriate.

Reviewer #1 (Remarks to the Author):

This research was novel and original, and I think it will interest others in the field.

This study suggests that Clec14a is a key endothelial-cell-specific target protein, and the effect of Clec14a genetic deletion was evident in accelerated trabecular bone formation and was closely linked to type-H vessel migration in the bone anlagen.

But there are some confused results of this work.

1) According to the previous study, in vitro investigations have shown that genetic deletion of Clec14a alters sprouting angiogenesis by diminishing EC sprout formation capacity downregulating EC tube formation (Clec14a genetically interacts with Etv2 and Vegf signaling during vasculogenesis and angiogenesis in zebrafish. *BMC Dev Biol.* 2019;19(1):6.), and genetic deletion of Clec14a in vivo, also resulted in compromised angiogenesis responses where vascular invasion into subcutaneously implanted sponges was impaired and accompanied by a diminished vascular density (Blocking CLEC14A-MMRN2 binding inhibits sprouting angiogenesis and tumor growth. *Oncogene*, 2015,19;34(47):5821-31.), those both proved the importance of Clec14a for type H blood vessels. In this study, Clec14a^{-/-} in vivo results in a decline of the type-H vascular front, but at the same time, accelerated trabecular bone formation. Please explain how type H vessels are linked to osteogenesis and the molecular mechanism.

Response:

The reviewer correctly notes that type-H vessels support migration of osteoprogenitors to sites of bone formation. Specifically, type-H capillaries have been shown to be crucial for postnatal osteogenesis (Kusumbe *et al.* 2014). The number of these vessels rapidly declines postnatally as bone formation rate decreases and, as a testament to the important role of these vessels for driving bone formation, an artificial increase in type-H vessel number in aged mice has been shown to increase bone formation and bone mass (Kusumbe *et al.* 2014, Kusumbe *et al.* 2016). Given these previous findings, we can understand the reviewer's expectation that a reduction in H vessels would be expected to lead to reduced bone density and that this appears contrary to our findings. However, when comparing wildtype mice with Clec14a^{-/-} mice, our results are in fact as follows:

1. The type-H vessel density (type-H vessels/mm) at postnatal day (P)4 and 4 weeks-old does not differ between genotypes (wildtype vs Clec14a^{-/-}) (Supplementary Figure 2C and D).
2. The number of type-H endothelial cells (not vessels) is lower in Clec14a^{-/-} mice at 2 weeks old (Figure 2A). We contend that this is due to more rapid maturation of the type-H vascular bed in Clec14a^{-/-} resulting in premature shortening of the length of H vessels by 4 weeks old (Figure 2D-E), but not at 4 days old (Supplementary Figure 4C).

It is important in this context to note that H vessels have a dual role: They promote bone formation by providing a conduit for osteoblast migration to sites of bone formation (Kusumbe *et al.* 2014), whilst also negatively regulating osteoblast maturation to maintain osteoblasts in

an immature pro-migratory state (Bohm *et al.* 2019). Osteoblasts located further away from blood vessels in the bone are more mature compared to the *Osx*⁺ immature osteoblasts located close to type-H vessels (Bohm *et al.* 2019). We also find that osteoblasts in the diaphysis are situated further from the vasculature compared to those in the metaphysis (new Supplementary Figure 6).

Maintenance of migratory osteoprogenitors is vital during postnatal osteogenesis. Their migration to sites of bone formation is controlled by the initial advancement of the type-H vascular front (Bohm *et al.* 2019, Xu *et al.* 2018) followed by its reduction. This reduction occurs in a tightly controlled manner between 6 days and 4 weeks postnatally to drive skeletal maturation by 12-weeks in mice (Langen *et al.* 2017, Kusumbe *et al.* 2014).

This transient nature of the H-type vasculature is the reason that 2-week-old mice were chosen for flow cytometry experiments (Figure 2A-B) - identification of a population of H-vessel endothelial cells (ECs) by flow cytometry after this time is extremely challenging because of their very low number. In the case of the *Clec14a*^{-/-} mouse, our data indicate that the reduction in H-vessel length occurs earlier than in the WT, giving a lower number of cells at 2 weeks of age. Our interpretation of these data is that skeletal maturation occurs more rapidly in the absence of *Clec14a*. This also explains the transient nature of the phenotype seen in these mice: Differences in bone volume are observed in mice during bone growth but not at peak bone mass or in aged mice (Figure 4).

In summary, we see a more rapid maturation of the vasculature in the bone of *Clec14a*^{-/-} mice. This manifests as a premature reduction in the length of the H-vessel compartment and an increase in the L vessel area but no difference in H vessel number. We find this to be entirely in keeping with the dense bone phenotype observed.

[FYI. All references referred to in the paragraph above are cited in full in the manuscript]

2) The evidence from this study confirms the relationship between *Clec14a* and type H vessels and osteogenesis. Still, there is insufficient evidence to show the signaling pathway through which *Clec14a* affects angiogenesis and osteogenesis, which needs to be further confirmed by additional data.

We agree. Reviewer 2 has suggested additional experimental work to enhance the evidence base, which we have addressed point-by-point below.

3) I don't think this paper will influence the thinking in the field.

We acknowledge the reviewer's view whilst noting their previous statement that: "...this research was novel and original, and I think it will interest others in the field." In response to reviewer 2's suggestions (below) we have now added additional data, which we trust will enhance the potential for our paper to influence thinking within this rapidly advancing field.

Reviewer #2 (Remarks to the Author):

In their manuscript with the title “Type-H endothelial cell protein Clec14a orchestrates osteoblast activity during trabecular bone formation and patterning”, Neag et al. report that C-type lectin domain containing 14A (Clec14A) is an important regulator of postnatal vessel growth in bone. Clec14a knockout mice display a reduction of bone vessels and, in particular, of type H capillaries, a subset of vessels that is associated with perivascular osteoprogenitors and is involved in the regulation of osteogenesis. The authors also show that skeletal development and osteoblast maturation are accelerated in postnatal Clec14a mutants. Further data indicates that the secreted protein Multimerin 2 (Mmrn2), which is highly expressed by type H endothelial cells, is involved in the communication with osteoblast lineage cells.

Overall, the presented findings are novel and interesting, but revisions are necessary to clarify several important issues.

Specific questions:

It is argued that Clec14a and Mmrn2 protein expression is associated with type H vessels in postnatal bone. The relevant stainings in Fig. 1, however, lack any endothelial marker. Thus, this claim seems not sufficiently supported by the data.

Thank you for this observation, we have now added additional data that strengthens our finding that Clec14a and Mmrn2 expression is limited to endothelial cells:

1. We have added single cell RNA sequencing data to the manuscript (new Figure 1C) that demonstrates mRNA level (transcriptional) expression of Clec14a and Mmrn2 exclusively on endothelial cells and absent from chondrocytes, fibroblasts, pericytes, osteoblasts and osteoblasts.
2. To further confirm Clec14a expression on H-type endothelial cells, we have performed additional staining of *in vivo* Clec14a-labelled bone sections with Cd31 and Emcn and have replaced the original figure part, which had Clec14a staining only, with a new panel that includes vascular markers (new Figure 1B).

Given that Mmrn2 is secreted, protein-level expression is less useful for identification of the cellular source than mRNA-level expression, which clearly demonstrates restriction to endothelial cells (new Figure 1C).

Likewise, it would be helpful to include a specific marker in support of the arterial localization of Mmrn2.

Evidence that the Mmrn2 staining indicated by arrowheads in Figure 1A is arterial comes from previously published work showing that veins and arteries are distinctly shaped and localised inside long bones in mice (See Kusumbe *et al.* 2014, Figure1 and extended Data Figure 2). In this context, we consider the existing staining to be sufficient. Full characterisation of the pattern of staining of Mmrn2 extends beyond the primary objectives of our current research which aims to study the role of Clec14a in type H vessels localised in a different area of the bone.

To clarify the source of our decision for the reader, we have added the following to the legend of Figure 1:

“White arrowheads indicate Mmrn2 staining on an arterial vessel (arterial vessel identified with reference to Kusumbe *et al.* 2014, Figure 1 and extended data Figure 2).”

It remains unclear how Clec14a is actually regulating angiogenesis in bone and some explanation is needed. Are EC proliferation or survival altered in the knockout mice?

Thanks to the reviewer's suggestions, we have now strengthened the argument in this manuscript that the Clec14a vasculature matures more rapidly. This is evidenced throughout Figure 2 by the reduction of the type-H vascular front (figure part C), increased blunting of the buds (F & G) and reduced filopodia (H) in the absence of Clec14a. Previous work has shown that Clec14a is crucial in controlling tip cell specification (Noy et al. 2017) and we show here that Clec14a is most highly expressed in the tip cells and at the site of vascular front advancement (Figure 1A).

We recognise the reviewer's query regarding a potential role for Clec14a in proliferation and/or survival of EC and have sought to address it. We have now added additional data (Supplementary Figure 2E) demonstrating that Clec14a does not affect endothelial cell apoptosis rate, indeed there is no evidence of apoptosis in any cell type at this timepoint in either genotype. Whilst identification of proliferating Osterix positive cells in tissue sections was relatively straightforward (and we now include these data in the manuscript - Supplementary Figure 6D), identification of proliferating endothelial cells proved challenging because we lack a nuclear endothelial cell-specific marker to colocalise with Ki67. The endothelial cell markers available to us (e.g., Emcn and Cd31) are expressed on the cell surface and do not allow for segmentation of individual endothelial cells. At their highest (2 weeks postnatally), the proportion of type H endothelial cells is low (<0.1% of total cells isolated from the bone, corresponding to ~10% of total endothelial cells) and decreases rapidly with increasing age such that post-weaning these cells make up a vanishingly small proportion of the total bone cells, making identification of the proliferating portion of these cells by flow cytometric analysis (with e.g., BrdU) extremely challenging.

Whilst we have therefore been unable to answer the reviewer's question directly, the lack of difference in vessel number at all timepoints leads us to the conclusion that there is no intrinsic defect in vessel formation in either genotype. We have added a sentence in the discussion to note that a limitation of this study is the lack of assessment of endothelial cell proliferation:

"In addition, a limitation of this study is a lack of data assessing the effect of Clec14a on endothelial cell proliferation rate."

Do type H cells lose their characteristic features?

We are unclear which characteristic features the reviewer is referring to and hope we have addressed the essence of their question in our response below.

Type H vessel morphology: Type-H endothelial cells are the cellular building blocks of type-H capillaries. H- and L-type vessels have distinct characteristic feature and can be distinguished based on their distinct location within the bone and by their morphology and structural organisation (Kusumbe *et al.* 2014). Type-H vessels are column-shaped capillaries located in the long bone metaphysis and terminate in anastomotic bulges near the caudal end of the growth plate. Conversely, type-L vessels have a sinusoid-like shape and form an extensively branched vascular network in the long bone diaphysis. As mice mature, type-H vessels lose their characteristic columnar appearance and the vascular network in the metaphysis becomes more similar in structure to that of the sinusoidal type-L vessels. We see this process occurring earlier in the absence of Clec14a^{-/-}.

Type H endothelial cell activity: Flow cytometry analyses have confirmed that the ratio of type-H to type-L vessels decreases as mice age/mature and bone formation rate reduces, which is circumstantial evidence of their pro-anabolic function (Langen *et al.* 2017). Endothelial cell coculture with mesenchymal stem cells (MSC) in spheroid experiments have shown that type-H EC indeed regulate MSC to osteoblast differentiation, when MSC are cultured with type-H ECs expression of osteoblast lineage markers in MSCs was reported; in contrast, type-L EC were unable to induce this change. Whilst these pro-osteogenic features of type-H ECs have thus been clearly demonstrated, research into how these cells change their functionality and features during postnatal bone maturation is hampered by their low number, which is further decreased from 2-weeks of age (peak type-H EC density), as mice reach skeletal maturation (Kusumbe *et al.* 2014, Xu *et al.* 2018). In our *Clec14a*^{-/-} model we have observed and reported blunting of type-H vessel buds at 4-weeks of age (Figure 2) - this is described in more detail below, in response to another of the reviewer's comments.

It is stated that type L sinusoidal vessels are not altered, but the representative images are not sufficiently good to support this claim. In particular, the diameter or branching pattern of sinusoidal vessels appears different in the images showing mutant samples.

In addressing the reviewers question we note that *Clec14a* expression is confined to the type-H vasculature and there is no detectable protein expression on type L vessels (Figure 1A). Given the role of type-H vessels in driving bone formation and the evident bone density phenotype present in the *Clec14a*^{-/-} mouse, the diaphysis and type L vasculature within it have not been the focus of our study.

We have checked the manuscript carefully to ensure that any statements about the L-type vessels are corroborated by our data and we can confirm that we make no statements that imply a lack of difference in their morphology or organisation in the presence or absence of *Clec14a*. The only statements made are those based on the data included e.g., Results, paragraph 3: "A significant decrease in type-H endothelium percentage was identified in 2-week-old *Clec14a*^{-/-} mice compared to *Clec14a*^{+/+} (wildtype - WT), whilst no difference in L-type endothelium (*Cd31*^{low} *Emcn*^{low}) was detected (Figure 2 B)."

The reviewer also includes the following comment in the "minor comments" section below, which we include here given its relevance to type L vessels:

***While it is not a focus of this study, weaker DAPI signal (see Suppl. Fig. 5) indicates potential changes in the *Clec14a* mutant bone marrow. Can you please comment?

We agree that any changes in L-type vasculature might be expected to influence bone marrow homeostasis, however detailed characterisation of the type-L vasculature (e.g., branch frequency or diameter) was beyond the scope of this manuscript.

To highlight this limitation to our manuscript, we have included the following statement in the discussion, paragraph 4:

"Of note, given the lack of expression of *Clec14a* on type-L vasculature, the morphology and organisation of the type-L vasculature was not studied in detail in this manuscript."

It is argued that loss of Clec14a changes vessel buds at the chondro-osseous interface. The images provided support some sort of change, but it would be very helpful to combine the vessel staining with a chondrocyte marker. Ultrastructural data would be useful or, if this cannot be obtained, higher magnification images.

Thank you for this suggestion. We have now added new and additional images to **Figure 2 (F-J)** showing higher magnification imaging of type-H vessels including the buds at the chondro-osseous interface (**F & H**) and the growth plate (**I-J**).

These additional images in Figure 2 show a blunting of type-H vessels buds in *Clec14a*^{-/-} compared to *Clec14a*^{+/+} (**F**), which we have quantified (**G**). We have also added higher magnification imaging to show the reduction in endothelial tip cell filipodia in *Clec14a*^{-/-} at the chondro-osseous interface (**H**).

The cellular patterning of chondrocytes in the growth plate was also investigated and our images indicate no patterning changes in respect to the arrangement of either types of chondrocytes or at the interface between type-H EC and chondrocytes (**I-J**).

The authors propose that osteoprogenitor maturation is accelerated in Clec14a mutants, but Osx is shown as the only marker. It would be important to look at Runx2+ cells.

We accept that additional evidence of accelerated osteoblast differentiation *in vivo* is desirable, however completion of additional time course experiments is no longer possible due to the absence of the *Clec14a*^{-/-} colony. We trust that the reviewer accepts our view that our data demonstrating the accelerated maturation of *Clec14a*^{-/-} osteoblasts *in vitro*, coupled with the RNASeq data and the *in vivo* phenotypic data all support our proposal. We have added a sentence to the discussion (paragraph 6) that reiterates this approach:

“Whilst detailed characterisation of osteoblast maturation state was challenging in *ex vivo* tissue sections, *in vitro* culture of calvarial osteoblasts freshly isolated from *Clec14a*^{-/-} mouse newborn pups also demonstrated increased propensity to differentiation....”

The calvaria data is interesting, but it is much less clear whether (and how) alterations in the vasculature and bone-forming cells cause this phenotype. This is a weakness and therefore this data is not adding much insight.

We agree that the calvaria data (original figure 4 G and H) is not central to the narrative and have therefore removed it from the manuscript.

Clec14a is presented as a negative regulator of osteogenic differentiation/maturation, but this is not entirely clear on the basis of the data provided. Can the authors check whether the association of osteoprogenitors and vessels is altered in Clec14a mutants, which might lead to partially uncoupled angiogenesis and osteogenesis. The data in Suppl. Fig. 5 is noted, but higher magnification images and quantification are needed.

We have added additional higher magnification images and quantification of the proximity of osteoblasts to vessels as a new Supplementary Figure 6. The quantification demonstrates that osteoblasts are located more closely to the vessels of the metaphysis than the diaphysis, as expected, but that this is not altered in the absence of Clec14a.

Minor comments:

The Discussion recapitulates a lot of aspects from the Results and could be trimmed or improved.

We agree with the reviewer and have trimmed the discussion to make it more succinct.

It is important to discuss the transient nature of the vessel defects in *Clec14a* mutants. Are there redundant gene products that get upregulated in adolescence or is *Clec14a* expression or function confined to growing vessels?

Clec14a expression is confined to type-H vessels. The transient nature of the phenotype is therefore entirely in keeping with the dramatic reduction in the type-H vascular compartment postnatally. We have added a sentence to the penultimate paragraph in the discussion to reinforce this point:

“Similarly, the effect of *Clec14a* genetic deletion on bone density was transient, resolving as bones reached skeletal maturity, in keeping with the transient nature of type-H vasculature in the bone anlagen.”

Fig. 3A: *Osx*⁺ cells in the diaphysis are presumably associated with the endosteum. Please clarify.

We have now clarified this in the methods section (3D immunofluorescent staining) by stating that:

“Osteoblasts proximal to the periosteum were excluded from the analysis for all samples.”

While it is not a focus of this study, weaker DAPI signal (see Suppl. Fig. 5) indicates potential changes in the *Clec14a* mutant bone marrow. Can you please comment?

We have addressed this comment above – see ***

Suppl. Fig. 5: It is argued that “*Osx*⁺ cells were also detected in the growth plate”. Presumably, this is *Osx* expression by chondrocytes? Please clarify.

Yes. We have now clarified this in the text by stating:

“A limited number of *Osx*⁺ cells were also detected in the chondrocytes of the growth plate (**Supplementary Figure 5**), as has been previously observed (4).”

Fig. 4G: Are control and mutant calvaria shown in the same anterior-posterior orientation?

These data have now been removed from the manuscript in response to the reviewer’s comment above.

The word “culture” is missing in line 4 of the paragraph beginning with “In vitro culture of calvarial...”.

Thank you. We have now corrected this in the text.

REVIEWERS' COMMENTS:

Reviewer #2 (Remarks to the Author):

Review of the revised manuscript COMMSBIO-23-4263A

First of all, I would like to thank the authors for their response to my comments and the inclusion of new (and better) images. Overall the manuscript is clearly improved, but I would ask the authors to carefully check and revise the wording of certain statements in the manuscript (see below).

The authors argue in their rebuttal letter that “Clec14a expression is confined to the type H vasculature”. This statement is not fully correct. First of all, only a tiny part of the distal diaphysis is shown in Fig. 1A, which obviously does not rule out that there is signal in the deeper bone marrow cavity. Second, not all the C2 (anti-Clec14a) signal in Fig. 1A is endothelial, which might reflect background noise or the existence of non-endothelial or perivascular Clec14a+ cells in the metaphysis. Furthermore, the transcriptomic analysis in Suppl. Fig. 1 supports that there is differential expression of Clec14a in H-type vs. L-type ECs, but expression is clearly not absent in the latter. Likewise, the scRNA-seq data in Fig. 1C do not show a clear difference among the majority of ECs shown in the tSNE blot. Thus, there is evidence for some level of Clec14a expression in the sinusoidal endothelium, which should be reflected in the wording of the text.

In their response to reviewer #1, the authors argue that type H vessels might negatively regulate osteoblast maturation, leading to more or faster differentiation. Furthermore, “acceleration of type-H vessel maturation” is mentioned in the Discussion as the cause of the changes in the Clec14a mutant metaphyseal vasculature (lines 267-271). Both points are indeed theoretical possibilities, which should be not be treated as established findings in the text in the absence of direct experimental evidence. For example, the data shown in the manuscript cannot rule out that Clec14a promotes vessel growth (including filopodia formation) at the chondro-osseous interface, consistent with previous studies showing that the transmembrane glycoprotein is a pro-angiogenic regulator. In fact, it would be highly beneficial to cover this point in a balanced discussion that covers different possibilities.

A final minor point is that Suppl. Fig. 4 is currently not mentioned anywhere in the text.